# Serial X-ray liquidography: multi-dimensional assay framework for exploring biomolecular structural dynamics with microgram quantities

Seong Ok Kim [1,2], So Ri Yun[1,2], Hyosub Lee [1,2], Junbeom Jo[1,2], Doo-Sik Ahn [1,2], Doyeong Kim [1,2], Irina Kosheleva[3], Robert Henning [3], Jungmin Kim[1,2], Changin Kim [1,2], Seyoung You [1,2], Hanui Kim[1,2], Sang Jin Lee [1,2] & Hyotcherl Ihee [1,2] ✉

Understanding protein structure and kinetics under physiological conditions is crucial for elucidating complex biological processes. While time-resolved (TR) techniques have advanced to track molecular actions, their practical application in biological reactions is often confined to reversible photoreactions within limited experimental parameters due to inefficient sample utilization and inflexibility of experimental setups. Here, we introduce serial X-ray liquidography (SXL), a technique that combines time-resolved X-ray liquidography with a fixed target of serially arranged microchambers. SXL breaks through the previously mentioned barriers, enabling microgram-scale TR studies of both irreversible and reversible reactions of even a non-photoactive protein. We demonstrate its versatility in studying a wide range of biological reactions, highlighting its potential as a flexible and multi-dimensional assay framework for kinetic and structural characterization. Leveraging X-ray free-electron lasers and micro-focused X-ray pulses promises further enhancements in both temporal resolution and minimizing sample quantity. SXL offers unprecedented insights into the structural and kinetic landscapes of molecular actions, paving the way for a deeper understanding of complex biological processes.

Proteins play vital roles, orchestrating life's fundamental functions including enzyme catalysis, signal transduction, and immune response. A wealth of techniques, from site-directed mutagenesis to high-throughput screening in diverse chemical libraries has enriched our understanding of protein functions, revealing regulatory pathways, unveiling protein-protein interactions, and identifying promising drug targets[1-5]. In addition, progress in structural biology using X-ray crystallography, nuclear magnetic resonance, and electron microscopy, has propelled our comprehension of protein structure and function, revealing intricate details about protein folding, conformational changes, and molecular interactions[6-8]. Yet, proteins are dynamic entities continuously shifting their shapes and interactions with other cellular components, as they perform their specific functions. To grasp the essence of the dynamic nature of proteins,

[1]Center for Advanced Reactions Dynamics (CARD), Institute for Basic Science (IBS), Daejeon 34141, Republic of Korea. [2]Department of Chemistry, Korea Advanced Institute of Science and Technology (KAIST), Daejeon 34141, Republic of Korea. [3]Center for Advanced Radiation Sources, The University of Chicago, 9700 South Cass Avenue, Argonne, IL 60439, USA. ✉e-mail: hyotcherl.ihee@kaist.ac.kr

understanding how their structures evolve over time is crucial. Despite employing various time-resolved (TR) approaches, capturing these fleeting changes during biological reactions remains time-consuming, resource-intensive, and laborious.

Unraveling the complexity of molecular interactions and orchestrated movements during biological reactions demands the ability to initiate reactions with exquisite precision and track their progression with the utmost temporal acuity. Traditional mixing methods offer a universal approach to trigger molecular reactions, but they typically demand a substantial sample quantity while offering only a modest temporal resolution[9–11]. On the other hand, laser-based techniques, particularly those using the pump-probe scheme, where a pump (optical) pulse initiates a reaction and a time-delayed probe (optical or X-ray) pulse monitors the progress of the reaction, have advanced the study of reactions by enabling initiation and precise tracking of reactions with high temporal resolution[12–17]. Nevertheless, while TR spectroscopy excels at probing fast reaction dynamics and spectral characteristics of intermediates, it lacks detailed structural information. TR X-ray structural techniques, such as TR X-ray crystallography (TRXC) and TR X-ray liquidography (TRXL), complement TR spectroscopy, as they can provide detailed structural information with remarkable spatiotemporal resolution, deepening our comprehension of molecular reactions[12–17].

Among these techniques, TRXL, which is also called TR X-ray solution scattering (TRXSS), was initially utilized to investigate the structural dynamics of small molecules in the liquid phase, and later its application was extended to probe real-time structural changes of biomolecules as well. When TRXL is applied to macromolecules, it is also referred to as TR small-angle/wide-angle X-ray scattering (TR-SAXS/TR-WAXS). TRXL experiments typically utilize an open jet sample delivery system when studying a small molecule solution[18–21]. Small molecules, often synthesized in large quantities at low cost, can be studied using open jet systems for irreversible reactions. However, this approach is impractical for most proteins because an open jet system requires large sample quantities, which are often difficult to obtain for proteins. For this reason, the application of TRXL to proteins has been mostly limited to studying photoactive proteins with fast, reversible photocycles. These proteins are ideal for a closed capillary system, which allows for repeated pump-and-probe measurements[22,23].

Furthermore, conventional TRXL has been largely limited to photoactive molecules, which constitute a small minority of biological molecules. This limitation arises because molecules lacking inherent absorption at commonly available wavelengths cannot be triggered by the pump laser pulses typically used in the pump-probe scheme of TRXL. Consequently, the application of TRXL is restricted to a small subset of biological molecules—those inherently photoactive. The use of photocaged molecules offers a promising solution. These molecules act as caged reaction initiators, releasing them upon photolysis with light[24–26]. By incorporating photocaged molecules, TRXL can potentially be extended to a much wider range of biological reactions, including those involving non-photoactive molecules. However, even with photocaged molecules, a significant challenge remains: many real-world biological reactions involve irreversible changes, leading to the sample consumption problem inherent in pump-probe measurements. This is because the used sample cannot be reused and must be discarded after a single measurement. Additionally, photocaged molecules are often costly, which can hinder their use in a typical pump-probe scheme, even if their corresponding biomolecules can be prepared in large quantities.

Conventional TR setups demand large sample quantities (i.e., sometimes several milligrams per time delay, totaling more than tens of milligrams of sample quantities for a time series with multiple time delays[25], see the Required sample quantities for irreversible reactions in conventional TRXL section of Supplementary Information) to collect a dataset with a reasonable signal-to-noise ratio (SNR) when

studying irreversible biological reactions[15]. Moreover, the situation is exacerbated when examining the kinetic behaviors of proteins under diverse environmental conditions, such as variations in ionic strength, inhibitors, pH, and temperature, since the demanding sample quantities increase in proportion to the number of parameters to be covered. Consequently, TR structural techniques in biological reactions have predominantly focused on reversible photoactive reactions under limited experimental conditions and cannot be widely employed to assay molecular reactions in biological systems. To overcome the challenge of limited sample quantities, an efficient sample utilization system is crucial for extracting structural and kinetic information from diverse biological reactions.

In this work, we introduce a versatile assay platform designed to explore the molecular reactions in the biological system using only microgram quantities of samples.

## Results

### Serial X-ray liquidography (SXL) platform
Traditional TRXL offers unparalleled advantages for characterizing molecular actions under near-physiological conditions while working with diverse sample types and experimental conditions without additional sample treatments such as crystallization, vitrification, or isotope (or chemical) labeling[12,14,15,26–29]. However, its significant drawback lies in sample utilization efficiency, especially detrimental for studying irreversible reactions, where sample reuse is impossible. To overcome this, we explored the possibility of employing a fixed target similar to the one utilized in serial femtosecond X-ray crystallography (SFX) to deliver the sample of interest[30–33]. While the initial fixed target design with interconnected microchambers proved impractical due to excessive liquid sample requirements, we devised an alternative approach by removing the connecting components, resulting in independent serially arranged reaction microchambers (Fig. 1, hereafter referred to as serial X-ray liquidography chip, shortly SXL chip). This simplified chip design, featuring an asymmetric layout with open and closed sides for each microchamber, and the truncated square pyramid microchambers, facilitates easy sample loading while maintaining the structural integrity of the chip (Fig. 1, Supplementary Figs. 1–3). Fabricated from polydimethylsiloxane (PDMS) with an organosilicon surfactant (Silwet L-77)[34] to ensure hydrophilicity, the SXL chip has 40 × 30 microchambers, each capable of being used for initiating and tracking molecular reactions (Fig. 1 and Supplementary Figs. 1–4). To guarantee pristine sample conditions throughout the data collection, the loaded SXL chip was sealed with a thin polypropylene film, providing low X-ray scattering and high optical clarity (Figs. 1, 2a, b and Supplementary Fig. 4). The SXL chip effectively preserved the integrity of the liquid sample for several hours, sufficiently long for collecting TR data (Supplementary Fig. 4h). Subsequently, we integrated the SXL chip with TRXL, referred to as serial X-ray liquidography (SXL), to study not only reversible but also irreversible reactions, and even the reaction of a non-photoactive protein.

### Validating the feasibility of the SXL method
While convenient for studying many photoactive reactions, conventional TRXL struggles to visualize irreversible biological reactions, such as enzyme-driven metabolism, protein degradation, and various posttranslational modifications. A key challenge associated with studying irreversible reactions is the high sample quantity required due to the irreversible nature of these reactions, preventing sample recycling and limiting data acquisition. Within these challenging targets, we especially focused on the irreversible molecular dissociation of *Arabidopsis thaliana* UV-B resistance 8 (*At*UVR8), a photoreceptor involved in plant stress responses triggered by harmful UV-B light[35,36], as a model system. Previous biochemical and structural studies reported that *At*UVR8 undergoes an irreversible dissociation from a homodimer (ground state) to monomers (signaling state) upon UV-B

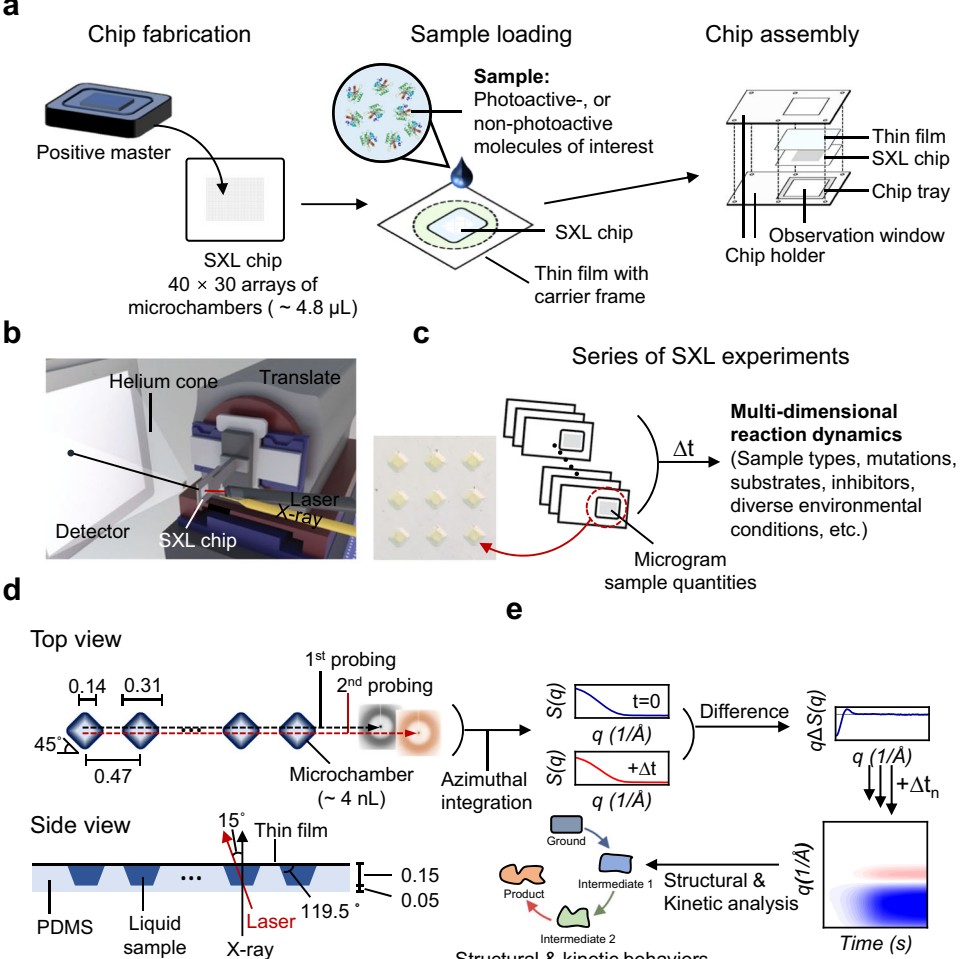

**Fig. 1 | Overview of the serial X-ray liquidography (SXL) method. a** Schematic of SXL chip preparation. (Left panel) The SXL chip was fabricated from a positive master. (Middle panel) The liquid sample was loaded onto the SXL chip. The hydrophilic nature of organosilicon surfactant and the design of the microchambers allowed for bubble-free loading. (Right panel) After the loading of the sample, the SXL chip was tightly sealed with a thin film and assembled with a sample holder for mounting on a translation stage. **b** Schematic representation of the beamline setup for SXL. **c** Schematic depiction of multi-dimensional reaction dynamics through a series of SXL experiments. An enlarged microscope image of the loaded microchambers is displayed on the left side. **d** (Top panel) Dual-probing schemes of the SXL method and dimension (in mm) of the microchamber from the top view. The serially positioned microchambers were designed to have truncated square pyramidal shapes and to have orientation rotated by 45 degrees with respect to the chip to enhance tolerance to x- and y-translation movement

uncertainties (approximately a 40% enhancement compared to the non-rotated configuration). The dual-probing scheme minimizes the sample waste by employing sequential X-ray pulses before (ground; 1st probing) and after (time-resolved; 2nd probing) the laser pulses from a single microchamber. (Bottom panel) Dimensions (in mm) of the SXL chip from the side view. Scattering curves before and after the laser pulse are integrated azimuthally. **e** A simple representation of data collection, along with structural and kinetic analysis. Scattering curves of the sample before (blue) and after (red) the laser pulses are collected, and the DS (difference scattering) curve between the scattering curves is obtained by subtracting the former from the latter. After collecting a series of DSs at different time delays, we generated two-dimensional contour maps to illustrate the reaction progress as a function of time. Structural and kinetic aspects of the biological reactions can be explored using a series of SXL experiments.

illumination[37–40]. This irreversible nature, combined with the limited sample quantities from purification, has hampered attempts to visualize the molecular actions of *At*UVR8 using conventional TRXL. We loaded an *At*UVR8 sample onto the SXL chip whose corresponding sample quantity was approximately 38 µg (0.1 mM, total volume approximately 4.8 µl). We employed 300 nm laser pulses to initiate the photoreaction, targeting the absorption band of Trp285, the main chromophore[36–38].

To validate the feasibility of our SXL method and assess its efficiency, we employed two data collection strategies: single- and dual-probing schemes (Supplementary Fig. 5). In the single-probing scheme where each microchamber is exposed to the X-ray pulse only once, yielding an X-ray scattering image, we acquired 40 pairs of scattering curves from two rows of microchambers (total 80 microchambers, approximately 32 ng for each microchamber; total sample quantity is

2.5 µg) out of 30 rows on an SXL chip, leaving 28 rows yet unused for subsequent measurements. Each pair of images comprised measurements without and with laser pulses, capturing both the ground and time-resolved states of the samples. Azimuthally integrating 2D images into 1D scattering curves and subtracting the scattering curves without the laser pulse from those with laser pulses yielded difference scattering curves (DSs), containing the information of structural change. Importantly, identical features across microchambers in both scattering curves and the DSs, confirm the uniform sample loading and integrity of the microchambers (Fig. 2c, d). While the single-probing scheme offers accurate and reliable TR data, it comes at the cost of substantial sample wastage. This is because it discards each sample portion after a single X-ray pulse, practically wasting half of the sample solely on collecting the reference static scattering signal needed for generating DSs. To address this, we devised the dual-probing scheme,

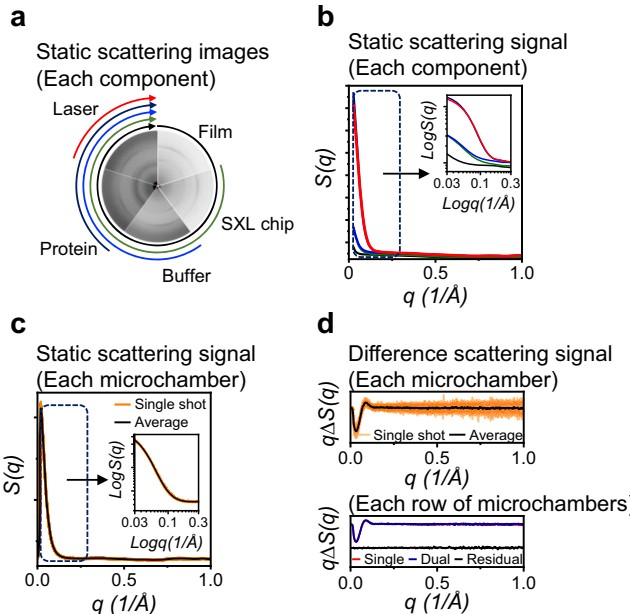

**a** Static scattering images (Each component)

**b** Static scattering signal (Each component)

**c** Static scattering signal (Each microchamber)

**d** Difference scattering signal (Each microchamber)

**Fig. 2 | Feasibility of the SXL method. a** Static scattering images presented clockwise from the top: film (black), a chip with film (green), a buffer-loaded chip with film (blue), a sample-loaded chip with film before (dark indigo) and after the arrival of a laser pulse (red). **b** Azimuthally integrated scattering curves of each component. Each curve is colored as indicated in (**a**). (Insert panel) Enlarged scattering curves are shown in a logarithmic scale. This plot indicates that SXL has a low background scattering. **c** Static scattering curves (without laser pulses) using the single-probing scheme. (Insert panel) Enlarged scattering curves in a logarithmic scale. **d** (Top panel) DSs at the time delay of 10 ms using the single-probing scheme. It is noticeable that both the static scattering curves and DSs show consistent features across different microchambers, confirming the integrity of the microchambers and their loaded samples. (Bottom panel) Comparison of DSs at the same time delay using the single- (red) and dual-probing (blue) scheme. Residual DS is shown in black. The obtained DSs from both the single- and dual-probing schemes exhibit identical features, indicating that subjecting the sample to two times of X-ray exposures did not result in noticeable sample damage.

which allows us to collect both reference and TR scattering signals from the same sample portion. We achieve this by utilizing sequential X-ray pulses, first without and then with a laser pulse for each microchamber (Fig. 1d and Supplementary Fig. 5). It is worth noting that the first X-ray pulse serves to probe the ground state of the sample before the reaction is initiated by a pump laser pulse. Subsequently, the laser pulse triggers the reaction and finally, a time-delayed X-ray pulse probes the time-resolved structural changes induced by the laser pulse. This approach enables us to utilize the sample twice as efficiently compared to the single-probing scheme. The two approaches (single- and dual-probing schemes) differ in the number of X-ray pulses used to probe, not the number of laser pulses. Importantly, the first X-ray pulse in the dual-probing method does not trigger the reaction. The laser pulse serves as the sole trigger for initiating the reaction.

To confirm sample stability after double exposure of X-ray pulses, we collected another pair of scattering curves from the next row of microchambers (total 40 microchambers, total sample quantity is approximately 1.3 μg) of the same chip used for the preceding single-probing scheme using the dual-probing scheme. Up to this point, three out of 30 rows were used. Comparing the DSs obtained from both the single- and dual-probing schemes showed that subjecting the sample to two times of X-ray exposures did not result in noticeable sample damage (Fig. 2e and Supplementary Fig. 6). Subsequently, we proceeded with pilot experiments on the remaining microchambers of the same chip (27 rows of microchambers, totaling 1080 microchambers) to assess the optimal experimental conditions such as laser fluence,

numbers of pulses per image, length of X-ray pulses train, and time range to be explored (Fig. 3a–d). From the pilot experiments using only a single chip, we could easily determine the optimal experimental conditions: 0.8 mJ/mm² of laser fluence, 40 microchambers per image, and 11 bunches of X-ray pulses ranging from 10 μs to 316 ms. These results were achieved using only a few micrograms of sample for determining each parameter, attesting to the high efficiency of SXL.

### In-depth application to an irreversible photoreaction

Following the optimization of experimental conditions for *At*UVR8, we applied SXL to explore its irreversible photoreaction. Employing another chip, we efficiently collected ten distinct time delays ranging from 10 μs to 316 ms, which correspond to three rows of microchambers (120 microchambers) per time delay (Fig. 4a). The SXL data shows a notable rise in the DS signal from 100 μs, followed by rapid propagation of a negative signal in the low-q region until 10 ms, and eventually reaching saturation after 31.6 ms (Supplementary Figs. 7, 8). On the other hand, the monomeric mutant (*At*UVR8-R146A/R286A), known to be inactive and incapable of photo-dissociation[37,38], displayed no detectable signals at all time delays (Fig. 4a and Supplementary Fig. 8). As a control experiment, we also conducted conventional static small angle X-ray scattering (SAXS) experiments to obtain the DS between the ground state of the wild type (dimer) and the monomeric mutant which mimics the active state of *At*UVR8. It is worth noting that the characteristics of the DS at the time delay of 316 ms from the SXL setup are consistent with that from the conventional static SAXS setup, confirming their remarkable agreement (Supplementary Fig. 9)[37,38]. Although a comprehensive structural analysis will be published elsewhere, we performed a preliminary kinetic analysis[15] using the singular value decomposition (SVD) and kinetics-constrained analysis (KCA) to validate the feasibility of SXL for capturing the reaction dynamics of *At*UVR8. By employing these methods, we extracted species-associated difference scattering curves (SADSs), which encapsulate the structural information of the corresponding reaction intermediates and determined the corresponding population changes of each species (Fig. 4b–d and Supplementary Fig. 10). While this kinetic analysis suggested that three key species exist during the photoreaction of *At*UVR8, it remains ambiguous to accurately determine the second time constant corresponding to the formation of the final product due to a low SNR and insufficiency in the number of time delays. Hence, to address this, we conducted two additional SXL experiments with a three-times higher sample concentration, 0.3 mM (115 μg of sample quantity), covering fifteen distinct time delays ranging from 31.6 μs to 100 ms, where the signal change was most drastic. This strategy led to a significant improvement in the SNR, allowing us to resolve the kinetic and structural features of each species (Fig. 4a). The SADS of the initial intermediate ($I_1$) exhibited subtle differences, suggesting a similar structure to the ground state with a slightly expanded outer shell, consistent with previous findings[40] (Fig. 4d). Subsequently, $I_1$ transformed into the second intermediate ($I_2$) with a time constant of $1.1 \pm 0.1$ ms. The SADS of $I_2$ resembles the DS from the static SAXS profiles between the dimer and the monomeric mutant (Fig. 4d and Supplementary Fig. 9), indicating that the conversion from $I_1$ to $I_2$ has characteristics of molecular dissociation. Finally, $I_2$ transitioned to the product (P) with a time constant of $13.4 \pm 2.8$ ms. While the structure of P resembled that of $I_2$, it was further characterized by an additional conformational change after molecular dissociation (Supplementary Fig. 11). The ability of SXL to deliver consistent datasets across repetitions (four datasets, considering two cycles of time delays from each chip, Fig. 4 and Supplementary Fig. 12) and concentration ranges (0.1 and 0.3 mM) with a different SXL chip underscores its remarkable fidelity in generating precise data, a testament to its transformative power in unlocking molecular events.

In a parallel effort to validate our SXL, we conducted a conventional TRXL experiment using a closed capillary setup, which required

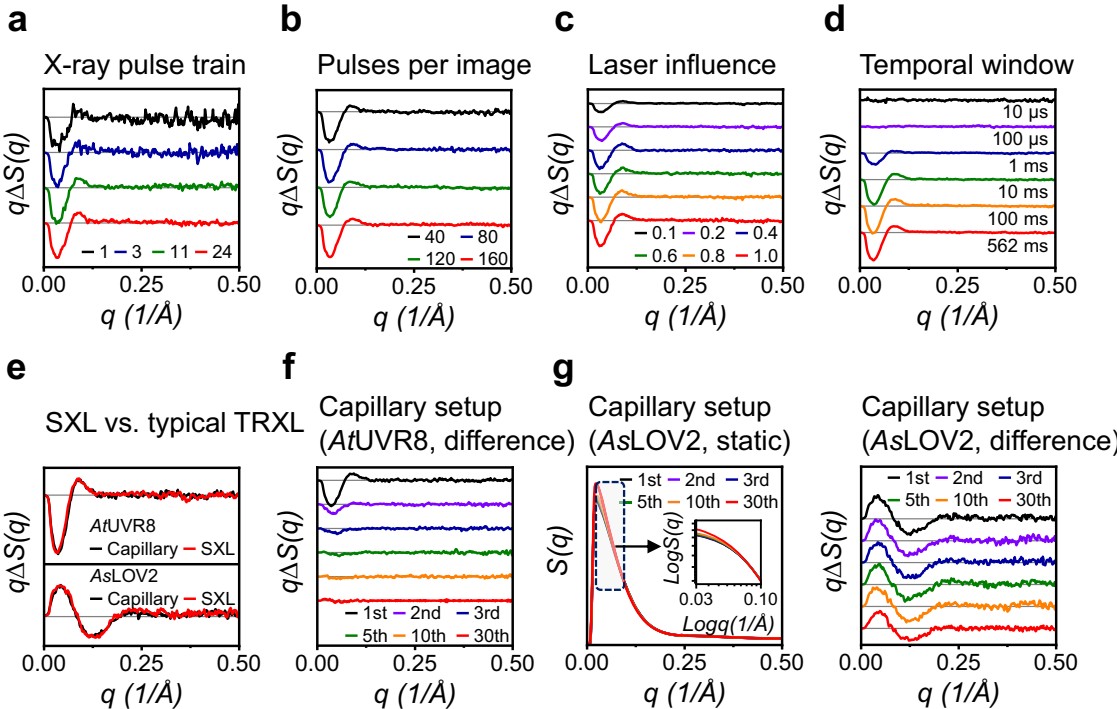

**Fig. 3 | Pilot experiments for optimizing experimental parameters and validation of the SXL method. a–c** DSs of *At*UVR8 at the time delay of 10 ms as a function of (**a**) the X-ray power, **b** the number of images (microchambers) per time delay, and **c** the laser fluence. **d** Screening the reaction time range for *At*UVR8 photoreaction. DSs from 10 µs to 562 ms are shown. Notably, using a few micrograms of sample quantities, we successfully optimized the experimental conditions (X-ray power, the number of images per time delay, laser fluence, and the time range to be explored) to be employed for the main data collection. **e** Comparison between DSs from SXL (Red) and conventional TRXL (Black). (Top panel) *At*UVR8. (Bottom panel) *As*LOV2. Notably, the first pair of measurements in TRXL data was

chosen to obtain a reliable DS. **f** Conventional TRXL data of *At*UVR8 at the time delay of 10 ms collected from the closed capillary as a function of the number of repetitions. The intensities and the shapes of the DSs changed rapidly due to the nature of the irreversible reaction. **g** (Left panel) Static scatting curves of *As*LOV2 from the closed capillary as a function of the number of repetitions. (Inset panel) Enlarged scattering curves in a logarithmic scale. The scattering curve changes rapidly in the Guinier region, suggesting that the photoreaction of *As*LOV2 eventually undergoes irreversibly. (Right panel) DSs of *As*LOV2 at the time delay of 10 ms as a function of number of repetitions.

a sample quantity of approximately 373 µg. The DS from the capillary was identical to that obtained from the SXL chip, confirming the compatibility of both approaches (Fig. 3e). However, in the capillary setup, the intensity of the DS signals rapidly diminished and their shapes changed significantly after a single measurement (Fig. 3f). In other words, while a single DS was obtained from the capillary setup using the sample quantity of approximately 373 µg, the SXL yielded the equivalent DS with only approximately 3.8 µg, demonstrating a sample utilization efficiency about 100 times greater than those of conventional setups.

**Applications to reversible photoreactions**

While the ability of SXL to capture molecular dissociation with microquantities of sample is impressive, questions remain about its sensitivity for unraveling intricate dynamics in molecules with smaller molecular weights or those exhibiting local structural changes. To address these concerns, we employed the light-oxygen-voltage sensing (LOV) domain 2 of *Avena Sativa* phototropin 1 (*As*LOV2), as a model system. *As*LOV2 undergoes light-induced conformational change within its Jα-helix and A'α-helix, but its structural transition of the photoreaction has not been studied by TRXL[41–47]. Using SXL, we successfully obtained 10-time delays (from 10 µs to 316 ms) of TRXL data from a single chip (Total sample quantity is approximately 169 µg, Fig. 5). Kinetic analysis revealed the major species and their corresponding time constants of 730 ± 60 µs and 8.6 ± 2.8 ms, aligning with previous spectroscopic studies (Fig. 5 and Supplementary Fig. 13)[45,47]. The SADS of the first intermediate (I₁) exhibited minor perturbations in the high q region, indicating local structural changes around the

chromophore as previously reported[47]. The SADS of the second intermediate (I₂) reflects a drastic conformational change occurring with a time constant of 730 µs, potentially due to the unfolding of the Jα-helix as suggested previously[45]. Interestingly, the SADS of the product (P) displayed oscillatory features in the high q region similar to that of I₂ but with significantly higher intensity in the Guinier region, indicating a more expanded structure. This raises the possibility of a dimeric state, in line with observations in other LOV domains[48–50] (Supplementary Fig. 14).

As a control experiment, we performed a conventional TRXL experiment using the closed capillary setup with a total sample quantity of approximately 390 µg, considering the reversible nature of the *As*LOV2 photoreaction[43,44,46]. While initially similar to SXL data, the DSs from the capillary setup rapidly degraded after only ten times of repetitions, losing their distinct features and becoming considerably weaker (Fig. 3e, g). Consequently, only four DSs retained sufficient quality for further analysis and exhibited good agreement with the SXL data set. Furthermore, the capillary setup exhibited concerning side effects with increasing measurements. First, the static scattering signal in the Guinier region rose rapidly, suggesting potential aggregation or sample damage (Fig. 3g). Secondly, the color of the sample in the capillary visibly shifted from fluorescent yellow to dark green (Supplementary Fig. 15). It demonstrates that *As*LOV2 is susceptible to cumulative damages from both X-ray and pump laser pulses, suggesting that even reversible reactions such as *As*LOV2 ultimately undergo irreversible damage under repeated measurements. These results visually highlight the superiority of SXL over the conventional setup,

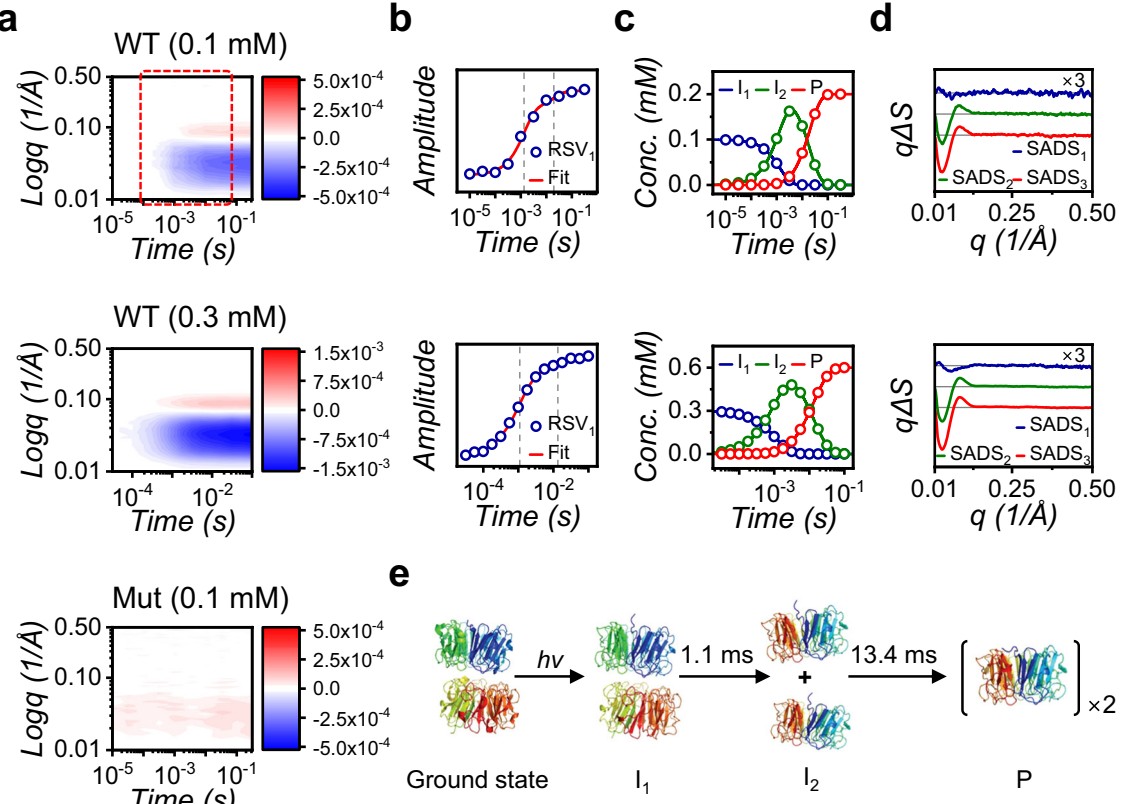

**Fig. 4 | Application of the SXL method to dissociative photoreactions of *At*UVR8. a** SXL data from *At*UVR8. (Top panel) Data from wild type at 0.1 mM (ten-time delays from 10 µs to 316 ms). This data identified the optimal time range (indicated in red square box) for data collection and the need for higher concentration. (Middle panel) Data from wild type at higher concentration (0.3 mM) and a focused time range (fifteen delays from 31.6 µs to 100 ms). (Bottom panel) The monomeric mutant (0.1 mM, ten-time delays from 10 µs to 316 ms). **b** Fits of the first RSV to obtain time constants of *At*UVR8 at 0.1 mM (top panel) and 0.3 mM (bottom panel) of sample concentration. **c** Obtained population changes of the intermediates of *At*UVR8 photoreaction at 0.1 mM (top panel) and 0.3 mM (bottom panel) of sample concentration. **d** SADSs for the reaction intermediates of *At*UVR8 photoreaction at 0.1 mM (top panel) and 0.3 mM (bottom panel) of sample concentration. It is noticeable that the time constants and structural information of intermediates from both the low-concentrated and high-concentrated samples match well, confirming the high reliability and accuracy of SXL. **e** Proposed reaction scheme for the wild type of *At*UVR8, derived from kinetic analysis of SXL data. While the wild type dissociates into monomers upon photoexcitation, the mutant, already monomeric in the dark state, remains intact after light exposure.

particularly for sensitive reversible reactions vulnerable to X-ray and laser damage.

Continuing our investigation, we sought to explore the feasibility of our SXL approach under diverse experimental conditions as a universal assay platform. This was previously hindered in typical TR techniques because of inefficient sample utilization and the lack of flexibility to adapt to various experimental conditions without laborious preliminary setups. We readily acquired SXL data for the I427V mutant of *As*LOV2 (*As*LOV2mut), known for its faster photocycle compared to the wild type[44], without any modification to the beamline setups or need for preliminary experiments. We determined the major species involved in the photoreactions and their corresponding time constants of $180 \pm 20$ µs, $4.2 \pm 0.7$ ms, along with identical SADSs to those of the wild type. These results revealed that the photoreactions of the wild type and mutant share similar structural properties throughout the reaction progression, but their kinetic properties differ (Fig. 5).

Next, we examined the kinetic behavior of the *As*LOV2 sample under various buffer conditions, including different external stimuli (laser pulse at the wavelength of 355 nm), high salt, low and high pH, and different concentrations of chemical denaturants. Utilizing the high efficiency and flexibility of our SXL approach, we seamlessly conducted these experiments without additional modification of the beamline setups or cleaning steps between the measurements (Fig. 5 and Supplementary Fig. 13). To evaluate the kinetic behaviors of *As*LOV2 samples under various conditions, we also conducted the brief kinetic analysis (Fig. 5). Although a further structural analysis regarding distinct intermediates for each condition is required to elucidate the molecular mechanisms of the photoreaction, a simple kinetic analysis provides the distinct features of the photoreaction in each condition (Supplementary Table 1): (i) The photoreaction triggered by a laser pulse at the wavelength 355 nm proceeds more rapidly than that at 450 nm. (ii) The formation of the product in a high salt condition is slightly slower than that in the isotonic condition, resulting in the recovery step being hardly observed within our temporal window. (iii) While the static scattering curve of the ground state in the low pH condition resembles that in the neutral pH condition, both the features of SADSs and the kinetic properties of signaling propagation differ significantly from those in the neutral pH condition. (iv) There is no significant difference in the photoreaction between the neutral and high pH conditions. Combining the results of (iii) and (iv), the photoreaction of *As*LOV2 is sensitive to the lower pH than the higher pH condition. (v) The kinetic and structural features of the photoreaction in 0.75 M Guanidine-HCl, often used to unfold biomolecular structures, appear to maintain their intrinsic native characteristics. This suggests that the globular structure of *As*LOV2 is resilient to such perturbed conditions. In addition, the second intermediate forms more rapidly, while the product forms more slowly than that of the wild type, indicating that 0.75 M Guanidine-HCl facilitates the unfolding process but interferes with intermolecular interaction (the formation of high oligomers). (vi) Finally, a distinct TR signal was not

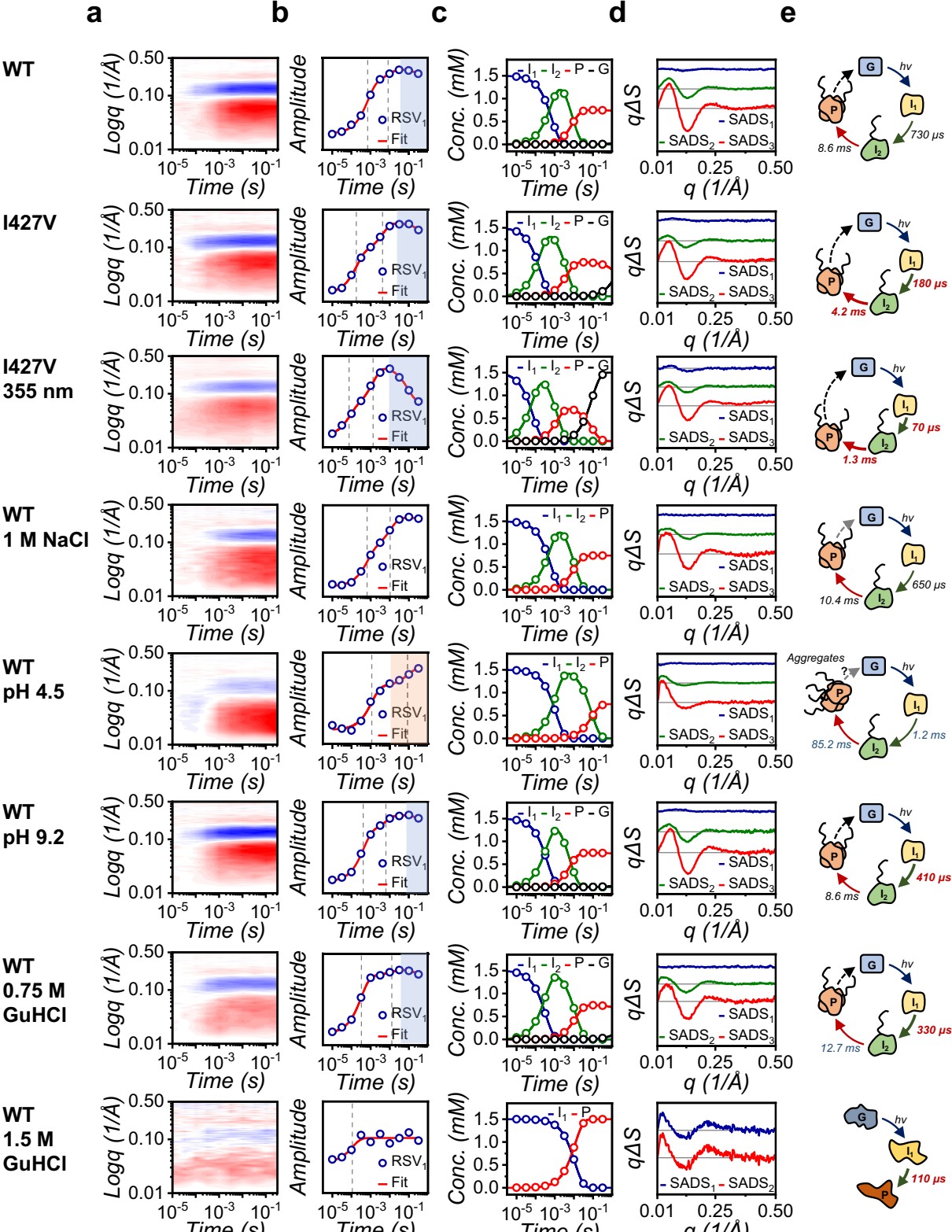

**Fig. 5 | Application of the SXL method to associative photoreactions of *As*LOV2.**
**a** SXL data from *As*LOV2 in diverse environmental conditions. **b** Fits of the first RSV to obtain time constants *As*LOV2 photoreaction in diverse environmental conditions. It is observed that the structural and kinetic aspects of *As*LOV2 in a low pH condition significantly differ from others, and the sample after the SXL experiment shows turbid-colored aggregates, suggesting instability of *As*LOV2 in acidic conditions. **c** Obtained population changes of intermediates in diverse environmental conditions. **d** SADSs for reaction intermediates in diverse environmental conditions. It is noticeable that, in high concentrations of NaCl and Guanidine-HCl, the recovery process is not observed. **e** Proposed reaction schemes from kinetic analysis of SXL data. The relative rates of species formation are indicated by arrow thickness and length: thicker and shorter arrows represent faster rates than those in photoreaction of the wild type, and vice versa.

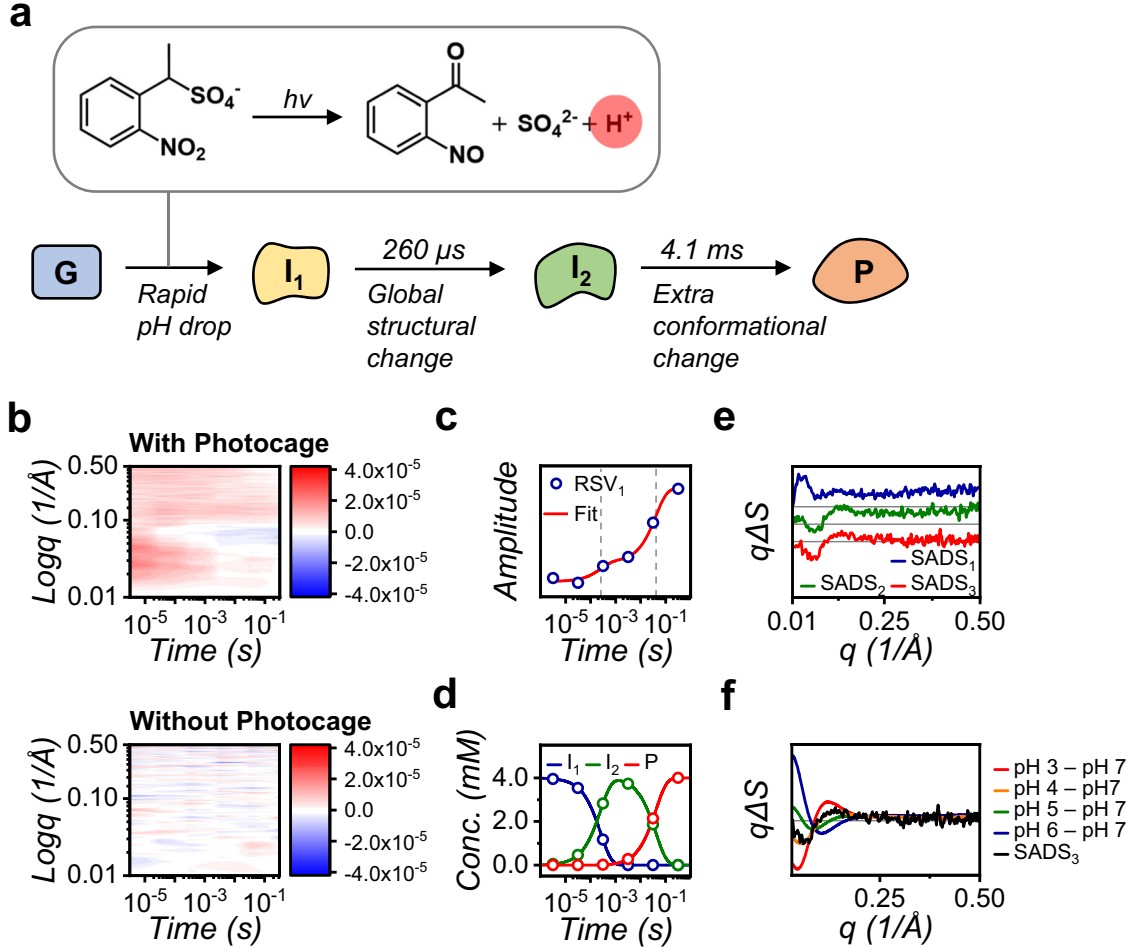

**Fig. 6 | Application of the SXL method to a non-photoactive biological reaction through the photolysis reaction of a photocaged compound. a** Proposed reaction scheme of the pH jump experiment on lysozyme. (Top panel) Molecular mechanism of the photolysis reaction of NPE-caged proton. (Bottom panel) Proposed reaction scheme from kinetic analysis of SXL data of lysozyme. **b** Two-dimensional contour maps of lysozyme reaction dynamics in the presence (Top panel) and absence (Bottom panel) of a photocaged compound. **c** Fits of the first RSV to obtain time constants of pH-jump SXL experiment. **d** Obtained population changes of intermediates in pH jump experiment. **e** SADSs for reaction intermediates of pH-jump from lysozyme. **f** Comparison of SADS3 (Product) and DSs obtained from conventional SAXS experiments. DSs were obtained by subtracting the SAXS signal of the designated pH from that of the neutral condition (pH 7.0). The features of SADS3 closely resemble that of the DS at pH 4.0, suggesting a pH reduction from neutral to 4.0 through the photolysis reaction of the photocaged molecule.

observed from the sample with 1.5 M Guanidine-HCl, suggesting that *As*LOV2 eventually loses its functional structure, becoming inactive at 1.5 M Guanidine-HCl.

Notably, collecting TR data covering such a broad range of sample conditions was completed in about two hours. This capability highlights the significant potential of the SXL approach as a multidimensional assay platform to screen diverse environmental conditions, mixtures of inhibitors, and various protein constructs, as well as complex compositions of molecular machinery for in-depth kinetic studies.

## Applications to a non-photoactive protein via a photocaged reaction

Pushing the boundaries of SXL, we explored its potential for visualizing the molecular actions of non-photoactive molecules using a photocaged molecule. We employed a photocaged molecule, 1-(2-nitrophenyl)ethyl sulfate (referred to as NPE-caged proton), which has the ability to induce significant pH changes and protonate groups with pKa values as low as 2.2[51,52]. As a demonstrative sample, we chose the hen egg white lysozyme, examining the structural and conformational change of the proteins under rapid pH change (Fig. 6 and Supplementary Fig. 16). Although the SXL data of the pH jump experiment on

lysozyme has much weaker signals than those of *At*UVR8 and *As*LOV2, we readily mitigated this by strategically adjusting the experimental setup. By increasing the number of measurements per time delay and reducing the total number of time delays from ten to six (ranging from 3.16 µs to 316 ms), we substantially enhance the SNR. The resulting kinetic analysis pinpointed the major species involved in the rapid pH change and their dynamic population changes. The obtained SADS of the first intermediate exhibited significant perturbations across the entire q range, indicating a rapid structural shift in lysozyme. The difference between the first and second SADSs suggested that local instability around the external shell ultimately leads to a global structural change. Furthermore, the SADS of the product (P) exhibited a distinct molecular shape compared to the neutral buffer conditions, aligning with the DS obtained between static scattering data from neutral and acidic conditions (Fig. 6d, e and Supplementary Fig. 17). While collecting static scattering data at various pH conditions using conventional SAXS required much larger sample amounts, it only provided a single structural information of the specific static state. In contrast, SXL delivers a comprehensive movie of the dynamic response, revealing the complex molecular motion of lysozyme during the rapid pH change.

## Discussion

The SXL method offers a significant advantage in terms of sample consumption compared to traditional techniques. For studying irreversible reactions, this benefit is particularly substantial. For example, for the irreversible photoreaction of AtUVR8, SXL requires only 115 μg of sample to obtain 30 DSs at 15-time delays (two DSs per time delay), translating to a mere 3.8 μg per DS. In contrast, conventional capillary setups require 373 μg of the sample to gather data at just a single time delay with one DS. Even for reversible reactions, SXL demonstrates the efficiency of sample utilization. Investigating the reversible reaction of AsLOV2 with SXL consumes only 169 μg of sample for 10-time delays with 30 DSs (3 DSs per time delay; 5.6 μg per DS), while 390 μg of the sample was consumed to obtain 30 DSs at a single time delay (10 ms), corresponding to 13 μg per DS. While the sample consumption of SXL and capillary setups becomes similar for 30 DSs (SXL: 5.6 μg vs. Capillary: 13 μg), a crucial factor comes into play. The AsLOV2 sample used in the capillary setup is susceptible to cumulative damage from both laser and X-ray pulses (Fig. 3g and Supplementary Fig. 15). This suggests the majority of the reversible reactions, like the AsLOV2 reaction, might suffer from irreversible damage during repeated measurements in the capillary setup. In the case of AsLOV2, only four DSs are usable from 390 μg, resulting in an effectively higher consumption of 97.5 μg per DS. Consequently, even for a reversible reaction, SXL maintains a clear advantage (effectively, SXL: 5.6 μg vs. Capillary: 97.5 μg).

Looking ahead, SXL promises to dramatically push the boundaries of TRXL with significantly reduced sample demands and high efficiency. Utilizing higher X-ray intensities, such as those available at XFEL beamlines will allow reliable data acquisition with a high SNR even from dilute samples. Micro-focused X-ray pulses dramatically increase sample utilization by reducing the required sample volume and quantity demands by orders of magnitude. This paradigm shift opens doors to previously inaccessible systems and complex processes requiring minimal sample quantities. While our synchrotron-based SXL experiments achieved microsecond temporal resolution, even faster timescales are within reach. By utilizing a single X-ray bunch instead of a train of multiple ones, the achievable resolution can be readily improved down to 100 picoseconds at the same synchrotron beamline[53]. Furthermore, at XFEL beamlines, this improvement can be significant, pushing the temporal resolution into the femtosecond regime. This notion is supported by the fact that even at the synchrotron beamline, the single-probing scheme, where a single X-ray image was obtained from a single X-ray pulse, provided reliable data with a sufficient SNR, suggesting the potential applicability of SXL to XFEL beamlines (Fig. 2c, d). SXL's versatility extends the ultrafast regime, excelling in studying slow reactions that occur over tens of seconds or minutes. The versatility of SXL stems from its well-crafted design, employing a series of independent microchambers that act as pristine reaction vessels. This configuration completely eliminates any risks of sample contamination from previous measurements, ensuring the integrity of each sample and the accuracy of TR data. In addition, SXL readily employs alternative data collection strategies such as the previously proposed method, hit-and-return system[54] to capture the reaction dynamics of slow reactions without requiring extra experimental setups.

In summary, we introduce the SXL method, which combines TRXL with a versatile and efficient sample delivery platform. By enabling diverse reaction studies with microgram sample quantities, SXL expands the frontiers of TRXL to previously inaccessible realms. Its combination of features, including independent microchambers, positions SXL as a versatile framework for unlocking the secrets of biomolecular dynamics across an even wider spectrum.

## Methods

### Sample preparation

The coding sequence of the LOV2 domain (residues 404–546) of phototropin 1 was amplified by PCR and cloned into a 2B-T vector (a gift from Scott Gradia; Addgene plasmid no. 29666) using ligation independent cloning (LIC) method. The AtUVR8 coding sequence (residues 12–381) was amplified by PCR and inserted into a pET22b vector (Novagen) using the NdeI and XhoI restriction sites, without a stop codon, to obtain the C-terminal His-tag (designated as pET22b-AtUVR8). The coding region of the C-terminal His-tag fused AtUVR8 was further amplified by PCR using pET22b-AtUVR8 plasmid and cloned into a 2M-T vector (a gift from Scott Gradia; Addgene plasmid no.29708) to generate the N-terminal MBP-His-tag and C-terminal His-tag fused AtUVR8 using LIC method (referred to as 2M-T-AtUVR8-His). The monomeric mutant of AtUVR8 (AtUVR8-R146A/R286A) and I427V mutant of LOV2 domain were generated using a Quickchange site-directed mutagenesis kit (Agilent) following the manufacturer's protocol. The plasmids used in the study are listed in Supplementary Table 3 and are available upon request.

To express the recombinant proteins, the cloned plasmids were transformed into Escherichia coli BL21(DE3) GOLD cells. The transformed cells were cultured overnight in Luria-Bertani (LB) broth and subsequently diluted 500-fold in 1 L fresh LB medium in a shaking flask. The cultivation temperature was kept at 37 °C until the optical density at 600 nm reached 0.6. Subsequently, the temperature of the shaking incubator was lowered to 18 °C, and the culture was allowed to cool down sufficiently for one hour. To induce protein expression, 1 ml of a 200 mM IPTG solution was added to each culture flask (final concentration of 0.2 mM), and the flasks were incubated overnight. The recombinant proteins were purified using IMAC Sepharose 6 Fast Flow (Cytiva) according to the manufacturer's instructions. After collecting the elution fractions of each sample, the N-terminal His-tag (AsLOV2 and its mutant) and the N-terminal MBP-His-tag (AtUVR8 and its mutant) were cleaved by TEV protease during the dialysis. The proteins were further purified using Q-Sepharose (Cytiva). For the AsLOV2 samples, the pooled fractions were subjected to IMAC Sepharose 6 Fast Flow to remove the trace impurities. To ensure the complete removal of the N-terminal MBP-His-tag and non-cleaved recombinant proteins from the pooled fractions of AtUVR8 and its mutant samples, the fractions containing the AtUVR8 and its mutant were passed through an MBPTrap HP (Cytiva) column and collected. The collected fractions were then dialyzed against 5 mM Tris, pH 7.0, 150 mM NaCl (AsLOV2 and its mutant) and 25 mM Tris, pH 7.5, 100 mM NaCl, 2 mM Dithiothreitol (AtUVR8 and its mutant).

All recombinant proteins are concentrated to 33.5 mg/ml (AsLOV2 and its mutant) and 24 mg/ml (AtUVR8 and its mutant) by Amicon® Ultra centrifugal filter units (Merck). To assay the reaction dynamics of AsLOV2 in diverse environmental conditions, we mixed the concentrated AsLOV2 samples with various additives stock solutions to make the desired conditions (See Supplementary Table 2). For the pH jump experiment, we purchased the hen egg white lysozyme (Merck) and NPE-caged proton (1-(2-nitrophenyl)ethyl sulfate; Tocris). Stock solutions of lysozyme (100 mg/ml) and NPE-caged proton (50 mM) were prepared by dissolving a proper amount of each molecule in deionized water with 100 mM NaCl (pH adjusted to 7.0 by using a concentrated NaOH solution). The final sample was made by mixing the stock solutions, resulting in a concentration of 60 mg/ml of lysozyme with 10 mM NPE-caged proton.

### Chip fabrication

The positive master was fabricated through a micro-milling process using ceramic filled PEEK (CMF) material. The positive master was designed with an additional dam architecture that is 50 μm higher than the height of the internal reverse part of the reaction microchambers. This design ensured that the thickness of the backside part of the serial scattering chip was 50 μm (Supplementary Fig. 1). To prepare the PDMS precursor (Dow Corning), the standard base was mixed with the crosslinker (w/w 10:1 ratio) and Silwet L77 (2% of the total volume of PDMS; PhytoTech Labs). These chemicals were thoroughly mixed using a

disposable plastic rod and subjected to vacuum to remove any trapped air bubbles. Both the positive master and the degassed PDMS precursor were preheated at 42 °C for 3 min. Subsequently, the mixed precursor was poured onto the positive master, and any excess amount of precursor was carefully removed using a slide glass (Supplementary Fig. 2). The positive master, covered with PDMS, was placed into a preheated 100 °C heat block for one hour. Afterward, it was cooled down to room temperature for 30 min. The cured PDMS membrane (SXL chip) was detached from the positive master using a scalpel.

## Sample loading

For sample loading, approximately 30 µL of the sample solution (Amount may vary depending on the sample viscosity) was pipetted onto the SXL chip. After waiting for 10–30 s to ensure the surface of the chip was adequately wetted, the sample was gently spread to cover the micro-arrayed reaction microchambers using a round-shaped capillary. To complete the sample loading process, any air bubbles present in the microchambers were gently removed by touching them with the capillary under a microscope. Once the air bubbles were removed, the excess sample solution was recovered through the capillary effect. To seal the chip, a thin polypropylene film (Chemplex Industries inc., SpectroMembrane® 3024) was placed over it. The sealed sample chip was then assembled with a sample holder and mounted on a translation stage for data collection.

## Beamline setup for SXL experiments

SXL measurements were performed at the BioCARS beamline of Advanced Photon Source (APS). X-ray focusing was achieved by using two pairs of Kirkpatrick-Baez mirror systems. The focused X-ray beam at the sample location had dimensions of $30 \times 30$ µm$^2$ (full-width-at-half-maximum, shortly FWHM). A high-speed Julich chopper delivered single isolated or multiple X-ray pulses, each with a pulse duration of 100 ps. To collect SXL data, 11 and 24 pulses were used, resulting in a temporal duration of 1.69 and 3.69 µs. The energy bandpass of the X-ray beam was defined by the beamline's upstream slits and, for this experiment, was 320 eV for an X-ray energy of 12 keV. In this setup, each isolated X-ray pulse delivered approximately $3 \times 10^9$ photons ($3.3 \times 10^{10}$ photons for 11 pulses and $7.2 \times 10^{10}$ photons for 24 pulses).

The reaction was initiated by a 7 ns laser pulse produced by a nanosecond OPOTEK laser at the chosen wavelength in the UV or visible range. We used 300 nm laser pulses (0.8 mJ/mm$^2$), 450 nm (1 mJ/mm$^2$), and 355 nm (10 mJ/mm$^2$) to initiate the photoreactions of AtUVR8, AsLOV2, and pH jump experiments of lysozyme, respectively. The laser pulse at the wavelength of 450 nm and 300 nm from the OPOTEK was transported through 100 µm optical fiber, and the laser pulse at the wavelength of 355 nm was delivered through optical minors to the final collimating and focusing optics system. It was then aligned in a geometry close to collinear with the X-ray pulse (approximately 15 degrees between the propagation directions of the laser and X-ray pulses). The laser spot at the sample location was approximately $100 \times 120$ µm$^2$ (FWHM). The time delay between the laser and X-ray pulses was controlled through a field-programmable gate array. A helium cone with a length of 360 mm was used to reduce air scattering. A Rayonix HS-340 detector was utilized to record the scattering images for every row (accumulation mode) or every microchamber (no accumulation mode).

## Data collection scheme for SXL experiments

To collect the SXL data, a raster scanning was used. The virtual grid plane of the raster scanning was defined by the positions of three points: the first point, which is the center position of the first microchamber in the top row; the second point, which is the center position of the last microchamber in the top row, and the last point is the center position of the last microchamber in the bottom row. The positions of these three points were aligned and defined using two orthogonal

viewing microscope cameras. The step size for this grid data collection was set to be equal to the distance between the centers of the microchambers. The grid was then moved through the beam by an ALIO diffractometer, ensuring that each microchamber of the grid was properly aligned with the X-ray and laser pulses. When collecting the SXL data using the single-probing scheme, each microchamber of the grid was exposed to the X-ray beam once. After collecting the scattering signals of the currently positioned row of microchambers, the grid was moved to the next row in a zigzag motion to replenish the microchambers for each measurement. In the dual-probing scheme, the laser pulse was blocked during the first probing step to acquire the scattering image of the ground state. After completing the first probing step, the ALIO diffractometer returned to the original position of the microchamber and the laser pulse triggered the reaction while the time-delayed X-ray pulse probed the reaction progress. Once the second probing step was finished, the measurement proceeded to the next row until the entire data collection was completed.

## Data collection for conventional TRXL and SXL experiments

In the conventional TRXL data collection, we used a closed capillary system using a Hamilton syringe (filled with approximately 50 µL of sample quantity) to load the sample into the capillaries. To minimize the vibrational jitter during ALIO diffractometer translation, we used capillaries with a diameter of 0.6 mm and a length of 55 mm. Conventional TRXL data of AsLOV2 and AtUVR8 were collected with a 1 Hz repetition rate at a time delay of 10 ms.

To collect the scattering signals of each component of the SXL platform, we first assembled the sample chip holder with a thin film and collected scattering signals. Then, we placed the empty chip on a thin film and assembled it with the chip holder to collect scattering signals of the chip and with the film (Fig. 2). Subsequently, we loaded the buffer solution into an empty chip and sealed it with a thin film to collect the scattering signals of a buffer-loaded chip with the film. Finally, we loaded the sample and collected the scattering signals of the sample-loaded chip with the film. To compare DSs from the single- and dual-probing scheme, we used three rows (120 reaction microchambers) of the AtUVR8 loaded chip, two for the single- and the other for the dual-probing schemes. To screen the optimal experimental parameters for the photoreaction of AtUVR8, we used the remaining 27 rows of microchambers with single usage. For the main data collection, we collected two pairs of DSs (fifteen-time delays from a chip) with a 2 Hz repetition rate. For the main data collection of AsLOV2, three pairs of DSs (ten-time delays from a chip) with a 2 Hz repetition rate. We also collected five pairs of DSs (6 time delays from a chip) from lysozyme with a 2 Hz repetition rate. All DSs were used for averaging and used for demonstration and further kinetic analysis. To evaluate radiation damage caused by multiple X-ray exposures, we utilized the flow-cell system with a capillary diameter of 0.6 mm. We loaded the AtUVR8 sample and collected scattering images repeatedly without laser pulses. In addition, we obtained scattering images of buffers using the same capillary to isolate the scattering signals from the pure protein sample.

## Data processing and kinetic analysis

The scattering curves were obtained by performing azimuthal integration of the scattering image by the FIT2D program (http://www.esrf.eu/computing/scientific/FIT2D/). The DSs were obtained by subtracting the scattering curves of the ground state from time-resolved scattering curves, as previously described[15]. The scattering curves of AtUVR8, AsLOV2, and lysozyme were normalized to the scattering signal in the range of $1.4 < q < 1.6$ Å$^{-1}$, following common practice and a previous report[28,29]. Any abnormal spike features in the scattering curves, resulting from detector noise, were removed during the post-data processing step using the Hampel filter incorporated in Matlab software. Singular value decomposition (SVD) analysis and

kinetics-constrained analysis (KCA) were performed using MatLab software, and the time constants were obtained using OriginLab software. To achieve the kinetic model of *At*UVR8, SVD analysis was applied to the $q$ range of 0.03 to 1.0 Å$^{-1}$ and the time range of 10 µs to 316 ms for low-concentrated sample (0.1 mM) and 31.6 µs to 100 ms for high-concentrated sample (0.3 mM). The first significant right singular vector (RSV) was fitted with a sum of two exponentials, resulting in time constants of $1.1 \pm 0.1$ ms and $13.4 \pm 2.8$ ms. Based on the previous biochemical studies and our results of the SVD analysis, a sequential kinetic model was applied as shown in Fig. 4. To extract the structural features of the transient species, SADSs were constructed using KCA analysis, as described in previous reports[15]. The theoretical DS at each time delay was generated by summing SADSs obeying the kinetics determined by the kinetic model. By minimizing the differences between the experimental and theoretical DSs at all time delays, we extracted the SADSs and obtained the time-dependent populations of each species. While constructing a kinetic model of *At*UVR8, we assumed that dissociation occurs during the transition from the first intermediate to the second intermediate, as described in the main text.

To obtain the kinetic model of *As*LOV2 and lysozyme, we also applied the SVD analysis as described previously. Based on the results of the SVD analysis and previous reports, we applied a sequential kinetic model. While constructing a kinetic model of *As*LOV2, we assumed that the third species (P) has a dimeric form, as described in the main text.

### Static scattering experiments

In addition to SXL, we also used the flow-cell system to collect the static scattering images of each sample as well as screen the initial experimental conditions such as pH, NaCl and sample concentration at the 4 C beamline of Pohang Light Source-II (PLS-II) at Pohang Accelerator Laboratory (PAL, Korea). To avoid the X-ray radiation damage, the static scattering images of each sample were obtained using the flow rate of 2 mm/sec, at which the scattering intensities did not show fluctuations caused by radiation damage. To validate the quality of each sample before the main data collection in the SXL mode, we collected the scattering images both without and with laser pulses at the BioCARS beamline of APS using the same flow-cell system.

### Reporting summary

Further information on research design is available in the Nature Portfolio Reporting Summary linked to this article.

## Data availability

The data supporting the feasibility of the SXL method for Figs. 2, 3 are provided in source data format. In-depth analysis of the data for Figs. 4–6, which demonstrate the general applicability of the SXL method but have not yet been fully analyzed at the structural level, is planned for a separate publication. To avoid redundancy and focus on key findings, we are not releasing the raw data publicly in source data format at this time. Instead, we are prepared to present processed graphical representations of the raw scattering profiles at various stages of reaction in the Supplementary Information. The datasets intended for the forthcoming publication, along with the complete original 2D scattering images, which are too extensive to include as source data, can be obtained from the corresponding author upon request. Source data are provided with this paper.

## Code availability

In this study, we conducted a brief kinetic analysis of each reaction using built-in plugin functions in OriginLab (for exponential fitting) and MATLAB (for SVD analysis). These functions can be utilized with the manufacturer's manuals. The KCA codes used in this study are available from the corresponding author upon request.

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

## Acknowledgements

We thank the 4 C beamline staff at Pohang Accelerator Laboratory (PAL, Korea) for assistance with SAXS data collection. This work was supported by the Institute for Basic Science (IBS-R033). The use of BioCARS Sector 14 was supported by the National Institute of General Medical Sciences of the National Institutes of Health under grant number P41 GM118217. Time-resolved set-up at Sector 14 was funded in part through a collaboration with Philip Anfinrud (NIH/NIDDK). We also thank Prof. M. Schmidt (University of Wisconsin-Milwaukee) for supporting the nanosecond laser system for time-resolved experiments at the BioCARS beamline.

## Author contributions

S.O.K., D.A. and H.I. conceived the idea and developed the method. S.O.K., J.J. and D.A. designed and fabricated the S.X.L. chip. S.O.K. screened and selected the target samples, and S.O.K., S.R.Y. prepared the samples. I.K., R.H. and S.O.K. set up the beamline and developed the data collection schemes. S.O.K., J.J., S.R.Y., H.L., D.K., D.A., C.K., S.Y., H.K. and S.L. participated in the data collection. S.O.K., S.R.Y., H.L., J.K., and D.K. processed data. S.O.K., S.R.Y., H.L. and D.K. analyzed the data. S.O.K. and H.I. wrote and edited the original drafts. HI supervised the whole project. All authors read and commented on the manuscript.

## Competing interests
The authors declare no competing interests.
