## [Peer Review File · Nature Communications]

REVIEWER COMMENTS

Reviewer #1 (Remarks to the Author):

The manuscript by Kim et al. describes a fixed target system of serially arranged microchambers to conduct solution scattering experiments. The authors demonstrate the utility of their system by measuring laser induced time-resolved scattering curves of a LOV domain, a photoreceptor involved in plant stress responses and hen egg lysozyme. These systems represent reversible, non-reversible and non-photoactive proteins, respectively, covering many potential use cases. In addition, the authors demonstrate how changing variables like pH, ionic strength or presence of desaturating agents would allow for multidimensional studies.

Solution scattering techniques like small angle X-ray scattering are established techniques to provide low resolution envelopes of protein structures and, in some cases, how the overall shape can change over time. The motivation for the development of the new sample delivery chip described in the manuscript was to lower sample consumption – this is in principle very desirable in any structural biology experiment, X-ray solution scattering, X-ray crystallography or cryo-EM both static and dynamic.

The manuscript emphasizes that an important application of the chips is to follow irreversible reactions. Although this might be true, the authors struggle to explain well to a reader (especially a non-expert one) why this is the case. Furthermore, it does not provide enough reference to published work, only general basically uninformative statements are provided e.g. Conventional TR setups demand ten to hundreds of sample quantities more than other biochemical assay methods, often exceeding what is readily available.

Furthermore, the authors' argument for efficiency of the proposed technique seems confusing. On one hand, the authors argue that the irreversibility of the reaction prevents samples from being re-measured, on the other they argue that it is the X-ray exposure (single vs double exposure scheme as explained in the article). This part is poorly explained.

My greatest concern is that the authors, in my opinion, fail to explain why the proposed method is more efficient and less sample consumption heavy as compared to previously used techniques (such as capillary setups). The whole term X-ray liquidography is confusing, as it only complicates comparison to the rest of the literature. Why not just use SAXS/WAXS or, if it is something new, explain the difference to the reader? How does the system, for example, compare to the high-throughput solution scattering beamline at DESY? Here sample amounts between 5 and 400 ug are needed for an automatic measurement (Blanchet et al., 2015), not so different than the sample

amounts stated in this manuscript. The authors just state general sentences like “For instance, for studying an irreversible reaction, TRXL typically requires tens or even hundreds of milligrams of protein samples to collect a dataset with a reasonable signal-to-noise ratio (SNR).” without providing references. The system is often compared to crystallography but perhaps a better discussion of other solution scattering studies would be more appropriate.

In the introduction and the discussion, the authors write repeatedly how ground-breaking their system is. I would like to see some examples of questions that can be answered that could not be answered before. They mention solution scattering at the XFEL or next generation synchrotrons would open a completely different time-domain allowing to answer novel questions. The manuscript would definitely deserve publication in Nature Communications if they had actually shown this and answered a new question. But in any case, the results should be compared to what had been done before. A classic example is the study of a photoreaction center at the LCLS by time-resolved wide-angle X-ray scattering (Arnlund et al., Nature Methods, 2014). What advantage would the described new solution scattering chips provide?

In conclusion the original content of this publication comprises the combination of a new sample delivery technique, that, similarly, is also used in time-resolved serial crystallography (for example Carrillo et al., IUCRJ, 2023), to X-ray solution scattering. While I am not an expert in solution scattering, the presented results seem sound, also taken into consideration that the experiments have been done with well-studied model proteins. In my opinion, the manuscript falls short in the explanation why the method is superior to similar techniques and what new science could now be done that was not possible before. It is up to the editor to decide if such a methodological step provides enough novelty for publication in a high-ranking journal. In any case, I would suggest changing the term X-ray liquidography, it is just confusing and I don't see the need to change the name of existing techniques.

Reviewer #2 (Remarks to the Author):

The object of any time-resolved experiment is generally the identification of kinetic or structural mechanisms that evolve in a crystal and the characterization and evolution of the population of states that comprise those mechanisms. Several groups have begun to develop technologies that try to explore the boundaries of what is measurable at both XFEL and synchrotron sources – the development of these new methods can be both complicated and challenging. The paper introduces a novel method which the authors call serial X-ray liquidography (SXL). In the paper the authors state that SXL overcomes current limitations of traditional time-resolved (TR) techniques in studying irreversible reactions and requiring large sample quantities. Therefore, by combining time-

resolved X-ray liquidography with a custom designed fixed target, enabling microgram-scale studies of both irreversible and reversible systems. The paper demonstrates the versatility of SXL and highlights its potential for kinetic characterization. The paper also briefly outlines the challenges in studying protein dynamics and the limitations of existing TR techniques. The paper is primarily a methods paper, introducing the SXL platform. The feasibility of SXL is validated through experiments on model systems, AtUVR8 and AsLOV2. The authors use of SXL demonstrates high efficiency and flexibility while offering insights into their dynamics.

Overall, the paper is clear and well written, the concept is sound, the data are convincing, and this is another novel another piece of technology with the potential to become useful in the long-term as the field of time-resolved protein science grows. Therefore, assuming my comments below can be answered, I would recommend publication in Nature Communications.

Since, the desired measured time-resolution should dictate the choice of radiation source. Are there limitations to this setup at other microfocus beamlines, since only a few are capable of utilizing single bunches using pink beams? Given the wider availability and use of microfocus beamlines at synchrotrons, their setup can become available to a larger array of users. What are the advantages and disadvantages. I am missing this discussion.

Could other materials be used for the fabrication process of the chips, such as COC or COP, as this has been used for the manufacturing of other polymer chips (<https://doi.org/10.1107/S2052252523007595>)? Why did the authors decide on PDMS?

How was the integrity of the microchambers ensured during sample loading and data collection? Especially if the authors plan on using such a platform at an XFEL. Given ultrashort pulses and the high photon density, it is possible that the features of these fixed targets could break. Do the authors plan on performing multi-shots per feature, will it be single-shot, or will the beam be attenuated when used at a FEL?

Were there differences between the setups for each of the model systems studied, if so, can the authors elaborate on the specific experimental setups and parameters used?

Given that the polymer film is placed on a carrier frame, how much curvature remains in the chip? Since any bowing will result in a misalignment when going from feature to feature.

Responses to the comments from Reviewer #1

The manuscript by Kim et al. describes a fixed target system of serially arranged microchambers to conduct solution scattering experiments. The authors demonstrate the utility of their system by measuring laser induced time-resolved scattering curves of a LOV domain, a photoreceptor involved in plant stress responses and hen egg lysozyme. These systems represent reversible, non-reversible and non-photoactive proteins, respectively, covering many potential use cases. In addition, the authors demonstrate how changing variables like pH, ionic strength or presence of desaturating agents would allow for multidimensional studies.

Solution scattering techniques like small angle X-ray scattering are established techniques to provide low resolution envelopes of protein structures and, in some cases, how the overall shape can change over time. The motivation for the development of the new sample delivery chip described in the manuscript was to lower sample consumption – this is in principle very desirable in any structural biology experiment, X-ray solution scattering, X-ray crystallography or cryo-EM both static and dynamic.

→ We appreciate the reviewer's constructive comments. We acknowledge the reviewer's insight into the significance of minimizing the required sample quantities for structural biology experiments. This aligns closely with the main objective of our SXL method, which aims to explore the structural transition of biological molecules in a time-dependent manner using minimal sample quantities. However, we would like to emphasize that beyond efficient sample utilization, the SXL method also offers versatility in studying diverse biological reactions. This flexibility positions SXL as a powerful multi-dimensional assay framework for kinetic and structural characterization of biological reactions. We have carefully considered the reviewer's comments and have addressed each point raised. In instances where the reviewer included unrelated multiple comments within a paragraph or raised the same concerns repeatedly in multiple paragraphs, we have reorganized our response to address them by topic for improved clarity. Below, we provide our responses to the reviewer's comments.

The manuscript emphasizes that an important application of the chips is to follow irreversible reactions. Although this might be true, the authors struggle to explain well to a reader (especially a non-expert one) why this is the case.

→ In response to the reviewer's feedback, we have carefully rewritten the manuscript to emphasize the importance of studying irreversible reactions and the significance of the SXL method in this context. We believe these revisions will enhance the manuscript's accessibility to broad scientific readers, encompassing both researchers with expertise in time-resolved methods and those with a more general scientific background.

The following is the revised part of the manuscript according to the reviewer's comments. The parts with major changes are indicated in bold.

“[...omitted...] Traditional mixing methods offer a universal approach to trigger molecular reactions, but they typically demand a substantial sample quantity while providing only a modest temporal resolution⁹⁻¹¹. On the other hand, **laser-based techniques, particularly those using the pump-probe scheme, where a pump (optical) pulse initiates a reaction and a time-delayed probe (optical or X-ray) pulse monitors the progress of the reaction**, have advanced the study of reactions by enabling initiation and precise tracking of reactions with high temporal resolution¹²⁻¹⁷. Nevertheless, while TR spectroscopy excels at probing fast reaction dynamics and spectral characteristics of intermediates, it lacks detailed structural information. TR X-ray structural techniques, such as TR X-ray crystallography (TRXC) and TR X-ray liquidography (TRXL), **complement TR spectroscopy, as they can provide**

detailed structural information with remarkable spatiotemporal resolution, deepening our comprehension of molecular reactions^{12–15}.

Among these techniques, TRXL, which is also called TR X-ray solution scattering (TRXSS), was initially utilized to investigate the structural dynamics of small molecules in the liquid phase, and later its application was extended to probe real-time structural changes of biomolecules as well. When TRXL is applied to macromolecules, it is also referred to as TR small-angle/wide-angle X-ray scattering (TR-SAXS/TR-WAXS). TRXL experiments typically utilize an open jet sample delivery system when studying a small molecule solution^{18–21}. Small molecules, often synthesized in large quantities at low cost, can be studied using open jet systems for irreversible reactions. However, this approach is impractical for most proteins because an open jet system requires large sample quantities, which are often impractical to obtain for proteins. For this reason, the application of TRXL to proteins has been mostly limited to studying photoactive proteins with fast, reversible photocycles. These proteins are ideal for a closed capillary system, which allows for repeated pump-and-probe measurements^{22–23}.

Furthermore, conventional TRXL has been largely limited to photoactive molecules, which constitute a small minority of biological molecules. This limitation arises because molecules lacking inherent absorption at commonly available wavelengths cannot be triggered by the pump laser pulses typically used in the pump-probe scheme of TRXL. Consequently, the application of TRXL is restricted to a small subset of biological molecules—those inherently photoactive. The use of photocaged molecules offers a promising solution. These molecules act as caged reaction initiators, releasing them upon photolysis with light^{24–26}. By incorporating photocaged molecules, TRXL can potentially be extended to a much wider range of biological reactions, including those involving non-photoactive molecules. However, even with photocaged molecules, a significant challenge remains: many real-world biological reactions involve irreversible substrate changes. This creates the same sample consumption problem inherent to the pump-probe measurements. This is because the used sample cannot be reused and must be discarded after a single measurement. Additionally, photocaged molecules are often costly, which can hinder their use in a typical pump-probe scheme, even if their corresponding biomolecules can be prepared in large quantities.”

Furthermore, it does not provide enough reference to published work, only general basically uninformative statements are provided e.g. Conventional TR setups demand ten to hundreds of sample quantities more than other biochemical assay methods, often exceeding what is readily available.

→ We appreciate the reviewer’s comment. We have revised the manuscript to avoid the uninformative statement. We acknowledge that the comparison of sample requirements between conventional TR setups and biochemical assay methods is irrelevant to the context of our study. We have removed this comparison from the manuscript to make it more concise and clear. For the final revised text, please refer to our responses addressing another recurring comment raised by the reviewer in our response to the reviewer’s other comment (*The authors just state general sentences like “For instance, for studying an irreversible reaction, TRXL typically requires tens or even hundreds of milligrams of protein samples to collect a dataset with a reasonable signal-to-noise ratio (SNR).” without providing references.*).

Furthermore, the authors’ argument for efficiency of the proposed technique seems confusing. On one hand, the authors argue that the irreversibility of the reaction prevents samples from being re-measured, on the other they argue that it is the X-ray exposure (single vs double exposure scheme as explained in the article). This part is poorly explained.

→ We appreciate the reviewer's constructive comment as it provides us with a chance to clarify the corresponding point regarding our data collection scheme. In this study, we employed two types of data collection methods: single-probing and dual-probing. These methods differ in the number of X-ray pulses used to probe, not the number of laser pulses. Importantly, the first X-ray pulse in the dual-probing method does not trigger the reaction. The laser pulse serves as the sole trigger for initiating the reaction. While we have provided a detailed description in the manuscript (including the main text, methods sections, and figure captions) as well as richly illustrated with schematics in Supplementary Fig. S5, we acknowledge the reviewer's point and have further refined the text to explicitly clarify the distinction between the two methods and the roles of the laser and X-ray pulses.

The following is the revised part of the manuscript according to the reviewer's comments. The parts with major changes are indicated in bold.

“[...omitted...] To address this, we devised the dual-probing scheme, which allows us to collect both reference and TR scattering signals from the same sample portion. **We achieve this by utilizing sequential X-ray pulses, first without and then with a laser pulse for each microchamber (Fig. 1d and Supplementary Fig. 5). It is worth noting that the first X-ray pulse serves to probe the ground state of the sample before the reaction is initiated by a pump laser pulse. Subsequently, the laser pulse triggers the reaction and finally, a time-delayed X-ray pulse probes the time-resolved structural changes induced by the laser pulse.** This approach enables us to utilize the sample twice as efficiently compared to the single-probing scheme. **The two approaches (single- and dual-probing schemes) differ in the number of X-ray pulses used to probe, not the number of laser pulses. Importantly, the first X-ray pulse in the dual-probing method does not trigger the reaction. The laser pulse serves as the sole trigger for initiating the reaction.**”

My greatest concern is that the authors, in my opinion, fail to explain why the proposed method is more efficient and less sample consumption heavy as compared to previously used techniques (such as capillary setups).

→ We appreciate the reviewer's suggestion to emphasize the efficiency of our SXL method compared to previously used techniques. Initially, we described the sample consumption and made a comparison with the closed capillary system for each reaction separately, which made it difficult to emphasize this advantage. To better highlight the efficiency of SXL in terms of the sample quantity requirements compared to previously used techniques, we have reorganized the manuscript to merge these comparisons into a separate section in Discussion. Additionally, we have included a dedicated discussion in Supplementary Information comparing our SXL method with conventional and potential alternative approaches for studying irreversible reactions.

Throughout the manuscript, we have highlighted the reduced sample quantities required by SXL as a key advantage over conventional methods. For instance, studying the irreversible reaction of *AtUVR8* with SXL utilizes about 100 times less sample quantity (115 μg to collect data at 15 time delays with two difference scattering curves (DSs) per time delay; 3.8 μg for each DS) compared to the capillary setup (373 μg to collect data at a single time delay with one DS). Similarly, we used 169 μg of sample to study the reversible reaction of *AsLOV2* (10 time delays with three DSs per time delay; 5.6 μg per DS), while 390 μg of the sample was consumed to collect data at a single time delay (10 ms, 30 DSs; 13 μg per DS) using the capillary setup. SXL and capillary setups consume similar sample quantities for thirty DSs (SXL: 5.6 μg vs. capillary setup: 13 μg). However, it turns out that the *AsLOV2* sample is susceptible to cumulative damage from both laser and X-ray pulses (Figure 3g and Supplementary Figure 15), suggesting that the reversible reaction of *AsLOV2* undergoes irreversible damage under repeated measurements, and only a few DSs (About four DSs) are usable. Considering this, even for a reversible reaction, SXL still holds an advantage over conventional setups (SXL: 5.6 μg

vs. Capillary setup: Considering that only four usable DSs are obtained from 390 μg . Effectively, 97.5 μg per DS).

In addition, considering the high sample utilization and efficiency of the SXL method compared to previous methods, we highlight that we were able to screen the optimal experimental parameters—such as laser fluence, number of pulses per image, length of X-ray pulse trains, and time range to be explored—using a single chip (Fig. 3a to d). These results were achieved using only a few micrograms of sample for each parameter. Conducting such a pilot experiment with previous methods is difficult, as they require large sample quantities and significant effort and time to adapt the experimental conditions for screening. This demonstrates the high efficiency and significant sample-saving nature of the SXL method compared to previous methods.

The following is the revised part of the manuscript according to the reviewer's comments. "The SXL method offers a significant advantage in terms of sample consumption compared to traditional techniques. For studying irreversible reactions, this benefit is particularly substantial. For example, for the irreversible photoreaction of *At*UVR8, SXL requires only 115 μg of sample to obtain 30 DSs at 15 time delays (two DSs per time delay), translating to a mere 3.8 μg per DS. In contrast, conventional capillary setups require 373 μg of the sample to gather data at just a single time delay with one DS. Even for reversible reactions, SXL demonstrates the efficiency of sample utilization. Investigating the reversible reaction of *As*LOV2 with SXL consumes only 169 μg of sample for 10 time delays with 30 DSs (3 DSs per time delay; 5.6 μg per DS), while 390 μg of the sample was consumed to obtain 30 DSs at a single time delay (10 ms), corresponding to 13 μg per DS. While the sample consumption of SXL and capillary setups becomes similar for 30 DSs (SXL: 5.6 μg vs. Capillary: 13 μg), a crucial factor comes into play. The *As*LOV2 sample used in the capillary setup is susceptible to cumulative damage from both laser and X-ray pulses (Figure 3g and Supplementary Figure 15). This suggests the majority of the reversible reactions, like the *As*LOV2 reaction, might suffer from irreversible damage during repeated measurements in the capillary setup. In the case of *As*LOV2, only four DSs are usable from 390 μg , resulting in an effectively higher consumption of 97.5 μg per DS. Consequently, even for a reversible reaction, SXL maintains a clear advantage (effectively, SXL: 5.6 μg vs. Capillary: 97.5 μg)."

The whole term X-ray liquidography is confusing, as it only complicates comparison to the rest of the literature. Why not just use SAXS/WAXS or, if it is something new, explain the difference to the reader?

→ We appreciate the reviewer's comment. First, we added SAXS/WAXS to the revised manuscript as follows.

"Among these techniques, TRXL, which is also called TR X-ray solution scattering (TRXSS), was initially utilized to investigate the structural dynamics of small molecules in the liquid phase, and later its application was extended to probe real-time structural changes of biomolecules as well. When TRXL is applied to macromolecules, it is also referred to as TR small-angle/wide-angle X-ray scattering (TR-SAXS/TR-WAXS)."

It is understandable that some may feel hesitant to adopt the term "liquidography" given the prevalent usage of SAXS or WAXS in biochemical studies. Although the reviewer has raised concerns about the term "X-ray liquidography" used in our manuscript, we would like to clarify the reasons why we have chosen to use this term instead of "SAXS/WAXS." First of all, "SAXS/WAXS" suggested by the reviewer overlooks its broad application in studying both solid and liquid samples. It is important to note that SAXS/WAXS terminology does not inherently suggest a "liquid solution", and it is indeed widely used for studying solid samples as well. The reviewer's strong suggestion to use SAXS/WAXS seems to indicate a focus solely on the solution scattering of biological molecules. However, as stated in our manuscript, our platform is designed to study both biomolecules and small molecules, including

a photocaged compound. For small molecules in particular, the term SAXS/WAXS can be confusing and may mislead readers by diverting attention from the primary focus of our SXL method, which is designed to explore diverse reaction dynamics of liquid samples using a fixed target system. Furthermore, the term, “X-ray liquidography” has been widely accepted and utilized for decades to describe techniques aimed at obtaining X-ray diffraction (scattering) images from liquid samples, analogous to the commonly used term “X-ray crystallography” for crystal samples. We believe the term accurately represents our method and its historical context, as supported by numerous references in the literature (Ihee, *Acc. Chem. Res.* 2009; Ki et al., *Annu. Rev. Phys. Chem.*, 2017; Kim et al., *Nature*, 2020, etc.).

How does the system, for example, compare to the high-throughput solution scattering beamline at DESY? Here sample amounts between 5 and 400 µg are needed for an automatic measurement (Blanchet et al., 2015), not so different than the sample amounts stated in this manuscript.

→ We appreciate the reviewer mentioning this reference. Firstly, we would like to address the sample quantities mentioned by the reviewer. It is important to note that the sample quantities in the work referred to by the reviewer are not specifically for time-resolved studies but rather for static SAXS experiments. This can be found in the last paragraph of page 434 of this paper, where the authors mention the required sample quantities for each static SAXS experiment (“Samples are measured at different concentrations, typically between 0.5 and 10 mg/ml, to detect interaction between the solutes. Between 10 and 40 µl of solution is required for each measurement...”). Even for the sample with a static X-ray scattering measurement, which resembles the scattering signal of the reaction at the ground state in the SXL method, our current version of the SXL method—serving as a proof-of-concept experiment for the feasibility of fixed targets for time-resolved study—can potentially yield 1,200 static measurements. For instance, in the case of *AtUVR8*, we used only 95.8 ng of sample to collect a reliable static scattering signal from each reaction chamber (Please refer to Fig. 2c in our manuscript), covering the SAXS and WAXS regions of the scattering signal. This capability demonstrates the efficient data acquisition of our SXL system, offering an advantage over the cited work in terms of sample utilization. While the Blanchet et al.’s work provides valuable contributions to the high-throughput solution scattering technique, its focus distinctly differs from the research questions we address here. For this reason, we have concluded that detailed comparisons with the Blanchet et al.’s work would be redundant and potentially distract readers from the core advantages of the SXL method.

The authors just state general sentences like “For instance, for studying an irreversible reaction, TRXL typically requires tens or even hundreds of milligrams of protein samples to collect a dataset with a reasonable signal-to-noise ratio (SNR).” without providing references.

→ We appreciate the reviewer’s constructive feedback. Here, we address both the current comment and a related comment starting with “*Furthermore, it does not provide enough reference to published work, only general basically uninformative statements are provided*”).

First of all, we apologize for the lack of references or detailed explanations in the introduction section of the manuscript, which contains generally uninformative statements. We have revised the manuscript to make it more concise and clear, incorporating relevant references. Regarding the required sample quantities when studying irreversible reactions, the required sample quantities exhibit significant variations depending on the intensity of the TR signal from molecules of interest. Consequently, the necessary sample quantities may vary widely, being either less or much more than what we initially outlined. Hence, we have revised the manuscript to exclude explicit mention of sample quantities. Instead, we have included references and provided a comprehensive discussion on the required amounts in Supplementary Information.

The following is the revised part of the manuscript according to the reviewer's comments. The parts with major changes are indicated in bold and the deleted parts are struck through.

~~“For instance, for studying an irreversible reaction, TRXL typically requires tens or even hundreds of milligrams of protein samples to collect a dataset with a reasonable signal to noise ratio (SNR). Conventional TR setups demand ten to hundreds of large sample quantities (i.e., sometimes several milligrams per time delay, totaling more than tens of milligrams of sample quantities for a time series with multiple time delays²⁵; See the “Required sample quantities for irreversible reactions in conventional TRXL” section of Supplementary Information)~~ **more than other biochemical assay methods, often exceeding what is readily available. to collect a dataset with a reasonable signal-to-noise ratio (SNR) when studying irreversible biological reactions¹⁵.”**

The following is the additional discussion of the required sample quantities when studying irreversible reactions in Supplementary Information according to the reviewer's comment.

“Required sample quantities for irreversible reactions in conventional TRXL

While specifying a required sample quantity for irreversible reactions can be challenging, large sample consumption is a well-recognized obstacle for studying such reactions with time-resolved methods. This is exemplified by a previous TR-SAXS study, where the molecular association of the nucleotide-binding domain of MsbA was investigated using the photocaged ATP molecule²⁵. In this study, the inherent nature of photolysis necessitated discarding the sample after each measurement. Based on the detailed description of sample usage provided in their Supporting Information, they utilized at least 45 milligrams of sample to collect TR-SAXS data at 13 time delays ranging from 50 ms to 1.4 s (3.5 mg per time delay). In comparison, for the irreversible photoreaction of *At*UVR8, SXL requires only 115 µg of sample to collect data at 15 time delays (7.7 µg per time delay). This estimation may underestimate the true sample consumption, as it only considers the sample used for the measurement and excludes additional waste generated by their flow-cell system, as well as the optimization of experimental conditions for data collection. While estimated sample usage may not accurately reflect the requirements of a general irreversible reaction, the substantial sample consumption observed in this study is a compelling illustration of the potential bottleneck this issue can pose, especially for studying diverse biological reactions, particularly irreversible ones. This is because biomolecule samples are often limited and difficult to prepare. As we have established throughout the text, one of the key advantages of SXL is its ability to utilize minimal sample quantities while still effectively exploring a wide range of biological reactions.”

The system is often compared to crystallography but perhaps a better discussion of other solution scattering studies would be more appropriate.

→ We appreciate the reviewer's constructive feedback. There seems to be a misunderstanding regarding the comparisons outlined in our manuscript. While the reviewer mentioned comparisons to crystallography, we focused on contrasting the SXL system with conventional TRXL methods (e.g., capillary-based, flow-cell systems) and alternative approaches. To avoid redundancy, we have presented the comparison of our SXL method to existing approaches (capillary setups) in the main text, and we have reorganized the manuscript to briefly compare the sample quantities used for both approaches in Discussion. In addition, a discussion section in Supplementary Information, titled "Drawbacks of conventional and potential alternative approaches to studying irreversible reactions," offers a more detailed comparison of SXL to conventional as well as alternative potential approaches (The detail of this section is given to our response to the reviewer's comment starting with *“But in any case, the results should be compared to what had been done before.”*).

In the introduction and the discussion, the authors write repeatedly how ground-breaking their system is. I would like to see some examples of questions that can be answered that could not be answered before. They mention solution scattering at the XFEL or next generation synchrotrons would open a completely different time-domain allowing to answer novel questions. The manuscript would definitely deserve publication in Nature Communications if they had actually shown this and answered a new question.

→ We appreciate the reviewer's feedback. This study focuses on introducing the SXL method, a time-resolved technique employing a fixed target system, specially designed to investigate diverse biological reactions in the liquid phase. These reactions, such as enzyme-driven metabolisms, protein degradation by protease, hydrolysis, various post-translational modifications (i.e., glycosylation, phosphorylation, and ubiquitination), DNA metabolism, molecular association (or aggregation)/dissociation, pose significant challenges for conventional TR structural biology techniques. A key challenge associated with studying irreversible reactions is the high sample quantity required due to the irreversible nature of these reactions, preventing sample recycling and limiting data acquisition. Within these challenging targets, we especially focused on the irreversible molecular dissociation of *AtUVR8*, a crucial biological process involved in regulating plant stress responses triggered by harmful UV light. This damaging radiation also affects numerous other living cells, including humans. We believe that this example precisely aligns with the reviewer's inquiry regarding "previously unanswered questions." To reflect the reviewer's point on the manuscript, we revised the relevant part as follows.

"Validating the feasibility of the SXL method

While convenient for studying many photoactive reactions, conventional TRXL struggles to visualize irreversible biological reactions, such as enzyme-driven metabolism, protein degradation, and various posttranslational modifications. A key challenge associated with studying irreversible reactions is the high sample quantity required due to the irreversible nature of these reactions, preventing sample recycling and limiting data acquisition. Within these challenging targets, we especially focused on the irreversible molecular dissociation of *Arabidopsis thaliana* UV-B resistance 8 (*AtUVR8*), a photoreceptor involved in plant stress responses triggered by harmful UV-B light^{35,36}, as a model system."

The significance of the current TR structural biology extends beyond elucidating the structure of individual biomolecules. It provides a window into understanding the complex mechanisms of vital biological reactions, ultimately paving the way for the development of valuable biocatalysts and other innovative applications such as biosensors and drug discovery strategies. However, applying TR structural biology approaches under diverse conditions has also been limited by high sample quantity requirements and complex, time-consuming setups that need modification for specific experiments. The SXL platform offers a user-friendly tool for exploring reactions of interest under various environmental conditions. As demonstrated in the manuscript, SXL allows easy transitions between different conditions without major modification of the experimental setups and the fixed target system. We anticipate that this universal flexibility of the SXL platform aligns well with the reviewer's comment by providing a solution as well as good examples that are not easily addressed by conventional methods. Combining these results, we believe these presented examples sufficiently demonstrate SXL's ability to address previously unanswered questions.

In addition, we carefully have revised the entire manuscript to avoid the overuse of terms like "ground-breaking." We focused on using more objective language to describe the significance and advantages of the SXL method, as follows. The deleted parts are indicated with lines struck through.

"Here, we introduce serial X-ray liquidography (SXL), a new technique that combines ~~innovative platform combining~~ time-resolved X-ray liquidography with a fixed target of serially arranged microchambers."

“In summary, we introduce the SXL method, ~~a ground-breaking method that~~ which combines TRXL with a versatile and efficient sample delivery platform. By enabling diverse reaction [···omitted···] unlocking the secrets of biomolecular dynamics across an even wider spectrum.”

But in any case, the results should be compared to what had been done before. A classic example is the study of a photoreaction center at the LCLS by time-resolved wide-angle X-ray scattering (Arnlund et al., Nature Methods, 2014). What advantage would the described new solution scattering chips provide?

→ We thank the reviewer for mentioning this relevant reference. It has prompted a useful comparison between our SXL method and past approaches, showcasing its potential benefits. The pump-driven microjet with a nozzle, as employed by Arnlund et al. (2014), was designed for ultrafast reactions. In terms of sample utilization efficiency, our SXL system demonstrates comparable sample consumption to the cited work. In their study, they used a flow rate of 10 $\mu\text{l}/\text{min}$ in 30 mg/ml of sample concentration with 120 Hz of repetition rate, resulting in an estimated sample consumption of at least 750 μg (83.3 ng is used for each measurement) per time delay. Our current SXL system, in its current demonstrative layout, consumes a similar sample quantity per measurement (As we described in the our response to the reviewer’s comment starting with “*How does the system, for example, compare to the high-throughput solution scattering...*”, 95.8 ng of sample quantity was used for each chamber when studying the irreversible reaction of *AtUVR8*). However, SXL offers a significant advantage in terms of future scalability and efficiency. Arnlund et al. employed a microfocus X-ray pulse of 10 μm , while our current setup utilizes that of 30 μm . By implementing a microfocus X-ray pulse similar to theirs, we can readily reduce the size of our microchambers by 70%, translating to a potential improvement in sample utilization efficiency by a factor of 3³ (By reducing the size of each dimension by two-thirds, one can achieve a 27-fold enhancement in sample-saving potential.), as SXL inherently allows for downsizing reaction chambers to match the X-ray pulse size.

Another key strength of the SXL platform is its adaptability to various timescales. Unlike the microjet method, which is severely limited in feasible observation timescales, SXL can readily adjust to collect a TR signal of reactions spanning moderate timescales such as microseconds to seconds—Many biological reactions occur in these timescales—without compromising sample utilization. In Arnlund et al.’s study, while the jet velocity of about 2.1 m/s is effective for studying the reaction dynamics of ultrafast reactions, it is not suitable for studying slower reactions due to a time-mismatch issue where the pumped sample portion moves away from the observable window before a time-delayed X-ray pulse reaches to probe. In their setup, the achievable longest time delay should be shorter than approximately 12 μs , even though they used a laser beam 25 times larger than that of the X-ray beam. Even neglecting the timing mismatch issue, such a microjet type of delivery system necessitates a larger sample quantity when collecting a TR signal at moderate timescales. For instance, acquiring TR data at the time delay of 100 ms would reduce the theoretical maximum repetition rate to 10 Hz, eventually decreasing the sample utilization efficiency to ~8% (= 10/120). In contrast, our SXL system readily adapts to such changes in experimental conditions without sacrificing sample utilization efficiency.

The following is the additional comparison of SXL to the cited work in Supplementary Information. The parts with major changes are indicated in bold. To save space and focus on the reviewer's comment, the first paragraph of the mentioned section, which is irrelevant, has been abbreviated with ellipses [···omitted···].

“Drawbacks of conventional and potential alternative approaches to studying irreversible reactions Existing methods such as closed capillary and flow-cell systems, along with droplet-on-demand (DOD) and liquid-jet approaches, were considered as potential [···omitted···] the flow-cell system requires more than a hundred times the sample quantity of the SXL approach, making it unsuitable for studying irreversible reactions.

Liquid-jet systems focused by an outer stream of inert gas, similar to flow-cell systems, have a high potential to save unwanted sample waste since these setups have been well established over a decade and often used to explore the reaction dynamics in the ultrafast timescale^{14,15,18,19,55}. However, these approaches share limitations that hinder their application to diverse biological reactions. While the high speed of the jet (i.e., repetition rate: 120 Hz, about 2.1 m/s⁵⁵) is effective for studying the reaction dynamics of ultrafast reactions, a liquid-jet system requires a significant sample quantity per time delay. In contrast, SXL has the potential to significantly enhance efficiency by reducing the reaction chamber size to match the X-ray pulse. As discussed previously, this reduction could proportionally increase efficiency by the cube of the size reduction. This highlights the potential of SXL for future optimization while demonstrating its comparable efficiency in its current layout. The situation becomes even more evident when studying slower reactions in the milliseconds time range. At slower timescales, a liquid-jet system requires a significantly reduced repetition rate and thus necessitates the consumption of a tremendously larger sample quantity. This limitation arises due to the inherent constraints on slowing down the microjet's sample delivery speed. For instance, acquiring data at a 100 ms time delay using the liquid-jet system used at an XFEL beamline—operated at a 120 Hz repetition rate to elucidate ultrafast reaction dynamics⁵⁵—would reduce the theoretical maximum repetition rate to 10 Hz, effectively decreasing the sample utilization efficiency to ~8% (= 10/120). In contrast, our SXL system readily adapts to such changes in experimental conditions without sacrificing sample utilization efficiency. This highlights the significant sample consumption required by the liquid-jet system at slower timescales.”

In conclusion the original content of this publication comprises the combination of a new sample delivery technique, that, similarly, is also used in time-resolved serial crystallography (for example Carrillo et al., IUCRJ, 2023), to X-ray solution scattering.

→ We are grateful to the reviewer for suggesting this pertinent reference (Carrillo et al., IUCRJ, 2023). To facilitate further comparison, we have included an additional discussion in Supplementary Information regarding the layout of a fixed target, comparing SXL with the reference.

While the layout of Carrillo et al.'s fixed target is designed for efficient protein crystal loading and shows promise for serial crystallography, it is deemed unsuitable when adapted for TRXL experiments. First, the edge structure near the center of the sample cell (pyramidal center apexes) can significantly affect background scattering patterns even with slight misalignment or fabrication imperfections of the fixed target. This makes obtaining precise TR signals challenging. Second, sealing the reaction chamber on both sides (one for sample loading and the other for the aperture) would be difficult, increasing the risk of liquid sample leakage. This potential weakness when applied to liquid samples poses severe challenges when rapid environmental changes occur during the reaction progress. Unlike this layout, our SXL method aims to observe reaction dynamics in various environments, including different salt concentrations, pH levels, temperatures, and the presence of various small molecules. In conclusion, while the SXL and Carrillo et al. systems share a similar layout, their inherent design caters to distinct functionalities and practical applications.

To address the reviewer's comment regarding the layout comparison, we have included a new discussion in the “Design and material selection for the SXL fixed target system” section of Supplementary Information.

“Design and material selection for the SXL fixed target system

Designing the SXL fixed target involved carefully considering how to handle liquid samples effectively. While our initial concept explored a microfluidic chip design known for its efficiency in capturing and

positioning crystals³¹, this approach was not suitable for SXL applications. In crystallography, the liquid acts as a carrier for the crystal sample of interest. In SXL, the liquid is the sample of interest, itself that needs to be preserved. Such a microfluidic design containing intra- and inter-connecting architectures requires a large liquid sample waste for the sample loading process. We then explored alternative fixed target designs that utilized discrete chambers for sample loading with no intra- and inter-connections^{30,32,33}. These both-side opened chamber layouts may avoid the unwanted sample waste for the sample loading process, but they presented challenges in efficiently loading the sample and maintaining the intact environments of the liquid sample during data collection. Moreover, sealing the reaction chambers on both sides (one for loading and the other for removing the excess liquid buffer) would also be difficult, increasing the risk of leakage. This potential weakness in maintaining the intact environment of liquid samples poses severe challenges when rapid environmental changes occur during reaction progress. To adapt the fixed target for the liquid sample, we removed the connecting microfluidic components while simultaneously closing one side of the reaction chamber. This modification ensured stable sample preservation for data collection. To enhance the sample loading process, we could consider a square pyramidal shape of reaction chambers, which was developed for serial crystallography³³. However, the edge structure near the center of the reaction chambers (pyramidal apexes) can significantly affect background scattering patterns, even with slight misalignment of the fixed target or fabrication imperfections. Therefore, we finally implemented truncated square pyramidal microchambers, having a flat surface on the bottom of the reaction chambers (100 μm \times 100 μm), as shown in Fig. 1.”

While I am not an expert in solution scattering, the presented results seem sound, also taken into consideration that the experiments have been done with well-studied model proteins. In my opinion, the manuscript falls short in the explanation why the method is superior to similar techniques and what new science could now be done that was not possible before. It is up to the editor to decide if such a methodological step provides enough novelty for publication in a high-ranking journal. In any case, I would suggest changing the term X-ray liquidography, it is just confusing and I don't see the need to change the name of existing techniques.

→ We sincerely appreciate the reviewer's constructive comments and positive recognition of our presented results. We have carefully considered the reviewer's feedback and have addressed each point. We also thoroughly reorganized the manuscript to more effectively present the benefits and superiority of our SXL compared to the conventional setups.

Responses to the comments from Reviewer #2

The object of any time-resolved experiment is generally the identification of kinetic or structural mechanisms that evolve in a crystal and the characterization and evolution of the population of states that comprise those mechanisms. Several groups have begun to develop technologies that try to explore the boundaries of what is measurable at both XFEL and synchrotron sources – the development of these new methods can be both complicated and challenging. The paper introduces a novel method which the authors call serial X-ray liquidography (SXL). In the paper the authors state that SXL overcomes current limitations of traditional time-resolved (TR) techniques in studying irreversible reactions and requiring large sample quantities. Therefore, by combining time-resolved X-ray liquidography with a custom designed fixed target, enabling microgram-scale studies of both irreversible and reversible systems. The paper demonstrates the versatility of SXL and highlights its potential for kinetic characterization. The paper also briefly outlines the challenges in studying protein dynamics and the limitations of existing TR techniques. The paper is primarily a methods paper, introducing the SXL

platform. The feasibility of SXL is validated through experiments on model systems, AtUVR8 and AsLOV2. The authors use of SXL demonstrates high efficiency and flexibility while offering insights into their dynamics.

Overall, the paper is clear and well written, the concept is sound, the data are convincing, and this is another novel another piece of technology with the potential to become useful in the long-term as the field of time-resolved protein science grows. Therefore, assuming my comments below can be answered, I would recommend publication in Nature Communications.

→ We appreciate the reviewer's thoughtful evaluation of our manuscript and constructive comments. The reviewer's recognition of the potential of our method in the field of time-resolved protein science is truly encouraging. We have carefully considered the reviewer's comments and have addressed each point raised. Below, we provide our responses.

Since, the desired measured time-resolution should dictate the choice of radiation source. Are there limitations to this setup at other microfocus beamlines, since only a few are capable of utilizing single bunches using pink beams? Given the wider availability and use of microfocus beamlines at synchrotrons, their setup can become available to a larger array of users. What are the advantages and disadvantages. I am missing this discussion.

→ We appreciate the reviewer's constructive comment regarding the applicability of our SXL method at other beamlines. We are confident that the SXL platform can be readily adapted to various beamlines without requiring significant modifications to existing experimental setups. While single-bunch pink beams excel at exploring ultrafast reaction dynamics, many biological reactions occur over moderate timescales ranging from microseconds to seconds or even hours. For these reactions, utilizing specialized ultrashort X-ray pulses is not essential. As discussed in the manuscript, SXL's adaptability extends beyond sample efficiency. The highly flexible design of our reaction chambers allows for adjustments to match the size of X-ray pulses at different beamlines. This flexibility, combined with efficient sample utilization, positions SXL as a universal tool for studying a wide range of biological processes across diverse timescales at various beamlines, including those with microfocus capabilities. Moreover, utilizing microfocus X-ray beams can further enhance sample utilization efficiency by reducing reaction chamber volume while accommodating a higher number of chambers.

In response to the reviewer's comment, we have included the following discussions in Supplementary Information. These discussions outline the advantages of utilizing SXL with a wide range of beamlines, including sample utilization enhancement (second paragraph) and flexible and efficient studies of diverse biological reactions across diverse beamlines (third paragraph). The parts with major changes are indicated in bold.

"Future perspective of the SXL method using alternative X-ray sources

In this work, we demonstrated the application of the SXL platform at a time-resolved beamline specialized for time-resolved Laue crystallography (14IDB of APS). We anticipate that utilizing alternative X-ray beamlines and facilities such as microfocus X-ray beamlines and XFELs holds great promise for future SXL applications, particularly with regards to enhancement of sample utilization (for both microfocus X-ray and XFEL beamlines) and reliable data acquisition from diluted samples with a high SNR and temporal resolution (for XFEL beamlines). The successful acquisition of the DS signal from a small sample volume demonstrated in this study will be a significant advance in the field of TR techniques. This capability opens up new possibilities for studying complex biological reactions, particularly those that involve scarce or precious samples.

The potential for further miniaturization of the microchambers, enabled by the use of a micro-focused X-ray beam, holds promise for even more efficient and high throughput data collection. In this submitted work, we successfully fabricated a microchamber with a volume of four nanoliters, precisely tailored to match the dimensions of the currently achievable spatially overlapped X-ray and laser pulses. With the use of a micro-focused X-ray beam, it is possible to further reduce the microchambers by more than several tens of times, reaching the scale of a few hundred picoliters. The current design of our microchambers allows for sufficient spacing between them, enabling the fabrication of a positive master through a simple micro-milling process. There is great potential to decrease the spacing between the microchambers, accommodating a higher number of microchambers to be accommodated within the same dimensions of the SXL chip. This modification facilitates the collection of a comprehensive data set in a single batch using a single chip without employing multiple chips for additional measurements. For example, by fabricating a microchamber at half the size of the current layout (top: 0.05 mm, bottom: 0.11 mm and height 0.075 mm, and spacing: 0.23 mm, resulting in a microchamber volume of 0.5 nL) within the same chip dimensions, we can collect 4,800 TR data points using just 2.4 μ L of sample. This demonstrates the potential to significantly increase the data acquisition capacity of our approach through the optimization of the microchambers and chip dimensions.

It is worth noting that a variety of biological reactions occur across a vast spectrum of timescales, ranging from microseconds to hours. For these reactions, highly specialized ultrashort X-ray pulses, which are less accessible to general users, are unnecessary. Instead, these reactions can be effectively investigated across a wider array of beamlines, including microfocus X-ray beamlines. As discussed in the main text, the SXL platform's adaptability extends beyond sample efficiency, and it can be readily adapted to various beamlines without requiring significant modifications to existing experimental setups in each beamline. This flexibility, combined with efficient sample utilization, positions SXL as a universal tool for studying a wide range of biological processes across diverse timescales at various beamlines. Moreover, even in scenarios where a pump-probe scheme is not available, slower reactions ranging from minutes to hours can be studied using SXL measurements with samples mixed with a reaction initiator or a photocaged molecule activated by a portable LED light source. Consequently, we anticipate that our SXL system can serve as a pivotal platform, enabling the study of diverse biological events for a broad spectrum of users through its efficiency enhancements and flexible capabilities.”

Could other materials be used for the fabrication process of the chips, such as COC or COP, as this has been used for the manufacturing of other polymer chips (<https://doi.org/10.1107/S2052252523007595>)? Why did the authors decide on PDMS?

→ We thank the reviewer for this constructive comment. Regarding the materials mentioned by the reviewer for the fabrication process of the SXL chip, we do not anticipate that these materials (COC or COP) meet the requirements for the application of SXL, as we explain below. In response to the reviewer's comment, we have included the following discussion in the “*Design and material selection for the SXL fixed target system*” section of Supplementary Information, which outlines the requirements for the application of SXL and explains the reasons for deciding on PDMS for the fabrication of the SXL chip. In addition, we also discuss the layout of the cited work with that of our SXL method.

“Design and material selection for the SXL fixed target system
[...omitted...]

Unlike typical fixed targets for serial crystallography that often utilize various materials such as silicon nitride, PDMS, and other polymers, the SXL chip is fabricated from PDMS. This choice prioritizes maintaining a perfectly homogeneous and intact environment for the liquid sample during measurements. Obtaining a clean background scattering signal is crucial for accurate TR signal

collection. PDMS fulfills these requirements for several reasons. PDMS's inherent adhesive nature facilitates a stable sealing environment when the thin film is pressed together, preventing leakage and maintaining sample integrity. Unlike other materials, PDMS allows for easy adjustment of its surface properties through various chemical treatments. By applying a surfactant to enhance the hydrophilicity of the PDMS and utilizing a truncated square pyramidal reaction chamber shape, we can facilitate the loading of aqueous samples. Furthermore, PDMS allows the SXL system to be applied in a wider range of solvent environments, extending its use beyond aqueous conditions. We envision its applicability to chemical reactions involving small molecules, where the liquid environments might be hydrophobic or aprotic. The easily controllable nature of PDMS hydrophobicity and hydrophilicity makes it ideal for such diverse applications. Moreover, PDMS offers a cost-effective solution with high precision and reproducibility. This is especially important for the SXL system, where repeated measurements are necessary to accumulate weak signals for difference scattering curve generation. PDMS ensures a stable and reliable measurement environment for consistent results. Studying reaction dynamics often involves using laser pumps, necessitating excellent optical clarity across a wide range of wavelengths, including UV light. PDMS provides outstanding transparency and stability across various wavelengths, allowing for clear observation of the reaction within the chip. PDMS exhibits high thermal stability, making it superior for tracking reaction dynamics across various temperature ranges. This is crucial for experiments studying biological responses to temperature changes or observing chemical reactions at different temperatures. On the contrary, other materials, especially cyclic olefin polymers, cannot meet the previously described requirements for the application of SXL to liquid samples. Therefore, PDMS, with its advantageous properties, was chosen as the optimal material for the SXL chip, ensuring a reliable and stable measurement environment for diverse applications.”

How was the integrity of the microchambers ensured during sample loading and data collection? Especially if the authors plan on using such a platform at an XFEL. Given ultrashort pulses and the high photon density, it is possible that the features of these fixed targets could break. Do the authors plan on performing multi-shots per feature, will it be single-shot, or will the beam be attenuated when used at a FEL?

→ We thank the reviewer for this constructive comment. We are confident in the SXL chip's ability to maintain the structural integrity of the microchambers throughout the experimental steps. Additionally, we believe that extending SXL to XFEL beamlines can be readily achieved without significant difficulty or the need for major modifications to the setups.

As discussed in the manuscript, the hydrophilic nature of the SXL chip and the wide-opening structure of the chamber enhance the sample loading process and prevent physical damage induced by capillaries. The microscope image of the sample-loaded SXL chip after data collection (Supplementary Fig. 4h) confirms the remarkable resilience of the chip to damage from radiation at third-generation synchrotrons. While this data supports the viability of SXL at synchrotrons, further testing under XFEL conditions is warranted. Although we acknowledge the potential damage to the backside of the PDMS in SXL chips during data collection due to ultrashort pulses with high photon density, the isolated reaction chambers within the SXL chip ensure that even if one chamber is damaged, subsequent measurements remain unaffected by simply moving to the next chamber. Moreover, the ultrashort pulse can probe the reaction dynamics of the molecules of interest before the microchamber is damaged (Chapman et al., 2011). For this purpose, a "single-shot" scheme to collect a TR dataset is necessary, which requires discarding the used sample after a single measurement. Again, the main concept of the SXL-single usage fixed target system for studying irreversible reactions, where it efficiently utilizes the sample quantity, comes into play. Finally, we do not expect additional attenuation to be necessary for

this collection scheme, since we can readily replenish the fresh sample content for subsequent measurements.

To address the structural integrity of the SXL chip and its feasibility for use at XFEL beamlines, we added these discussions to the “*Ensuring flatness and reliable handling in the SXL chip design*” section and “*Future perspective of the SXL method using alternative X-ray sources*” section in Supplementary Information, as follows. To save space and focus on the reviewer's comment, the first paragraph of the mentioned section, which is irrelevant, has been abbreviated with ellipses [...omitted...], and the parts with major changes are indicated in bold.

“Ensuring flatness and reliable handling in the SXL chip design

To achieve the flatness of our SXL chip, we explored various materials for both the film and structure of the sample holder system, ultimately leading to the [...omitted...] confirming the uniform sample loading, integrity, and flatness of the SXL chip.

As discussed in the main text, the hydrophilic nature of the SXL material (PDMS) and the wide-opening chamber design promote efficient sample loading. This design minimizes the risk of damage from capillary-induced physical contact during loading, as confirmed by the absence of observable damage from capillaries and the intact sample-loaded SXL chip after data collection (see Supplementary Fig. 4h). This feature ensures the chip's integrity and flatness, which are essential for obtaining reliable TR signals from the SXL chip.”

“Future perspective of the SXL method using alternative X-ray sources

In this work, we demonstrated the application of the SXL platform at a time-resolved beamline specialized [...omitted...] **efficiency enhancements and flexible capabilities.**

To expand the applicability of the SXL to XFEL beamlines, it is crucial to maintain the structural integrity of the SXL during data collection, especially under the intense ultrashort pulses and high photon flux characteristic of XFEL pulses. While the intense XFEL pulses raise concerns about damaging the SXL chip, its design offers inherent advantages. Even if a chamber is damaged by an XFEL pulse, the independent nature of the SXL chip's reaction chambers ensures unaffected subsequent measurements. We can simply move on to a fresh chamber within the chip using the single-probing scheme. In conclusion, the SXL chip's design and its compatibility with XFEL's "diffraction before destruction" principle⁵⁹ make it a promising candidate for future XFEL-based studies, offering superior temporal resolution and reduced sample consumption.”

Were there differences between the setups for each of the model systems studied, if so, can the authors elaborate on the specific experimental setups and parameters used?

→ We appreciate the reviewer for this question. The experimental setups remain identical, except for the setups employed to deliver pump optical pulses for different wavelengths and fluences in each target reaction. This consistency in experimental setups is beneficial, allowing the SXL system to observe reactions universally across various target systems, irrespective of wavelengths and sample environments. To improve the clarity of the experimental setups, we have split the paragraphs entitled “*Beamline setup for SXL experiments*” in Methods into two paragraphs: one explaining the probe X-ray pulse and the other describing the pump pulse, as follows.

“Beamline setup for SXL experiments

SXL measurements were performed at the BioCARS [...omitted...] delivered approximately 3×10^9 photons (3.3×10^{10} photons for 11 pulses and 7.2×10^{10} photons for 24 pulses).

The reaction was initiated by a 7 ns laser pulse produced [...omitted...] to record the scattering images for every row (accumulation mode) or every microchamber (no accumulation mode).”

The following is the revised part of the discussion in Supplementary Information according to the reviewer’s comments. The parts with major changes are indicated in bold.

“The SXL method empowers an efficient and straightforward exploration of biological reactions

Our use of consistent experimental setups, particularly the fixed target layout, for studying reaction dynamics across various demonstrative examples is noteworthy. Notably, only the laser pulse wavelength was varied, highlighting the flexible application of SXL across different targets.

The SXL method serves not only as a universal assay framework for studying diverse biological reactions by minimizing unwanted sample waste but also provides advantages over conventional methods in data collection.”

Given that the polymer film is placed on a carrier frame, how much curvature remains in the chip? Since any bowing will result in a misalignment when going from feature to feature.

→ We appreciate the valuable insight. During our initial examination of the composition of the SXL chip, we dedicated considerable attention to its flatness to ensure reliable data production. We appreciate the opportunity to emphasize this aspect. To address the reviewer’s comment on ensuring a flat and reliably handleable SXL chip design, we have added a detailed discussion titled “*Ensuring flatness and reliable handling in the SXL design*” in Supplementary Information. The following is the revised part of the manuscript according to the reviewer’s comments.

“Ensuring flatness and reliable handling in the SXL chip design

To achieve the flatness of our SXL chip, we explored various materials for both the film and structure of the sample holder system, ultimately leading to the utilization of a polymer film (Chemplex Industries Inc., SpectroMembrane® 3024) with a carrier frame. This specialized film has been widely used as a sealing material and X-ray transparent observation window for the sample cup in X-ray fluorescence measurements, ensuring a flat surface and reliable sealing properties. The polymer film with a carrier frame, featuring a square paper frame along the edges, effectively prevents the formation of wrinkles on the thin film. Consequently, the use of this film significantly reduces the likelihood of bubbles or wrinkles appearing on the thin film SXL chip during sample loading. Furthermore, by employing a rigid paper frame to maintain the flatness of the SXL chip after sample loading, we ensure ease of handling. Additionally, the sample chip holder, detailed in Supplementary Fig. 3, incorporates a chip tray structure whose dimensions match the size of the chip and are carved out to the thickness of the SXL chip from the surface of the plate, to prevent unwanted movement or flexing of the SXL within the holder. Finally, upon assembly of the upper and lower plates of the sample chip holder, the excess film and paper frame are removed, and the tight coupling between the plates ensures that the film remains flat, as demonstrated in Supplementary Fig. 4. As shown in Fig. 2c, the static scattering profiles from each reaction chamber within the whole chip, show identical features across microchambers, confirming the uniform sample loading, integrity and flatness of the SXL chip.”

REVIEWERS' COMMENTS

Reviewer #1 (Remarks to the Author):

I would like to congratulate the authors for the successful revision. My criticisms and suggestions have been well addressed in the point-to-point reply and have been incorporated into the revised manuscript. Readability and context to previous work have much improved. My recommendation is to continue with the editorial process towards publication. Looking forward to doing my own TR-SXL experiments one day.

Reviewer #2 (Remarks to the Author):

With the revised manuscript, the authors have addressed my comments and concerns.